# CAN SINGLE-PASS CONTRASTIVE LEARNING WORK FOR BOTH HOMOPHILIC AND HETEROPHILIC GRAPH?

## ABSTRACT

Existing graph contrastive learning (GCL) typically requires two forward pass for a single instance to construct the contrastive loss. Despite its remarkable success, it is unclear whether such a dual-pass design is (theoretically) necessary. Besides, the empirical results are hitherto limited to the homophilic graph benchmarks. Then a natural question arises: Can we design a method that works for both homophilic and heterophilic graphs with a performance guarantee? To answer this, we analyze the concentration property of features obtained by neighborhood aggregation on both homophilic and heterophilic graphs, introduce the single-pass graph contrastive learning loss based on the property, and provide performance guarantees of the minimizer of the loss on downstream tasks. As a direct consequence of our analysis, we implement the **S**ingle-**P**ass **G**raph **C**ontrastive **L**earning method (SP-GCL). Empirically, on 14 benchmark datasets with varying degrees of heterophily, the features learned by the SP-GCL can match or outperform existing strong baselines with significantly less computational overhead, which verifies the usefulness of our findings in real-world cases.

## 1 INTRODUCTION

Graph Neural Networks (GNNs) (Kipf & Welling, 2016a; Xu et al., 2018; Veličković et al., 2017; Hamilton et al., 2017) have demonstrated great power in various graph-related tasks, especially the problems centered around node representation learning, such as node classification (Kipf & Welling, 2016a), edge prediction (Kipf & Welling, 2016b), graph classification (Xu et al., 2018), etc. Prior studies posit that the good performance of GNNs largely attribute to the homophily nature of the graph data (Pei et al., 2020; Lim et al., 2021b; Zhu et al., 2020b; Abu-El-Haija et al., 2019; Chien et al., 2020; Li et al., 2021; Bo et al., 2021), *i.e.*, the linked nodes are likely from the same class in homophilic graphs, e.g. social network and citation networks (McPherson et al., 2001). In contrast, for heterophilic graphs, on which existing GNNs might suffer from performance drop (Pei et al., 2020; Chien et al., 2020; Zhu et al., 2020b), similar nodes are often far apart (e.g., the majority of people tend to connect with people of the opposite gender (Zhu et al., 2020b) in dating networks). As a remedy, researchers have attempted to design new GNNs able to generalize well on heterophilic graph data (Pei et al., 2020; Abu-El-Haija et al., 2019; Zhu et al., 2020a; Chien et al., 2020; Li et al., 2021; Bo et al., 2021).

For both homophilic and heterophilic graphs, GNNs, like other modern deep learning approaches, require a sufficient amount of labels for training to enjoy a decent performance, while the recent trend of the Graph Contrastive Learning (GCL) (Xie et al., 2021), as an approach for learning better representation without the demand of manual annotations, has attracted great attention. Existing work of GCL could be roughly divided into two categories according to whether or not a graph augmentation is employed. First, the *augmentation-based* GCL (You et al., 2020; Peng et al., 2020; Hassani & Khasahmadi, 2020; Zhu et al., 2021a;b; 2020d;c; Thakoor et al., 2021) follows the initial exploration of contrastive learning in the visual domain (Chen et al., 2020; He et al., 2020) and involves pre-specified graph augmentations (Zhu et al., 2021a); specifically, these methods encourage representations of the same node encoded from *two augmentation* views to contain as less information about the way the inputs are transformed as possible during training, *i.e.*, to be invariant to a set of manually specified transformations. Secondly, *augmentation-free* GCL (Lee et al., 2021; Xia et al., 2022) follows the recent bootsrapped framework (Grill et al., 2020) and constructs different

views through *two encoders* of different updating strategies and pushes together the representations of the same node/class.

In both categories, existing GCL methods typically require two graph forward-pass, *i.e.*, one forward-pass for each augmented graph in the augmentation-based GCL or one for each encoder in augmentation-free GCL. Unfortunately, theoretical analysis and empirical observation (Liu et al., 2022; Wang et al., 2022a) show that previous GCL methods tend to capture low-frequency information, which limit the success of those methods to the homophilic graphs. Therefore, in this paper, we ask the following question:

*Can one design a simple single-pass graph contrastive learning method effective on both homophilic and heterophilic graphs?*

We provide an affirmative answer to this question both theoretically and empirically. First, we theoretically analyze the neighborhood aggregation mechanism on a homophilic/heterophilic graph and present the concentration property of the obtained features. By exploiting such property, we introduce the single-pass graph contrastive loss and show its minimizer is equivalent to that of Matrix Factorization (MF) over the transformed graph where the edges are constructed based on the aggregated features. In turn, the transformed graph introduced conceptually is able to help us illustrate and derive the theoretical guarantee for the performance of the learned representations in the down-streaming node classification task.

To verify our theoretical findings, we introduce a direct implementation of our analysis, **S**ingle-**P**ass **G**raph **C**ontrastive **L**earning (SP-GCL). Experimental results show that SP-GCL achieves competitive performance on 8 homophilic graph benchmarks and outperforms state-of-the-art GCL algorithms on all 6 heterophilic graph benchmarks with a nontrivial margin. Besides, we analyze the computational complexity of SP-GCL and empirically demonstrate a significant reduction of computational overhead brought by SP-GCL. Coupling with extensive ablation studies, we verify that the conclusions derived from our theoretical analysis are feasible for real-world cases.

Our contribution could be summarized as:

- We show the concentration property of representations obtained by the neighborhood feature aggregation, which in turn inspires our novel single-pass graph contrastive learning loss. A directly consequence is a graph contrastive learning method, SP-GCL, without relying on graph augmentations.
- We provide the theoretical guarantee for the node embedding obtained by optimizing graph contrastive learning loss in the down-streaming node classification task.
- Experimental results show that without complex designs, compared with SOTA GCL methods, SP-GCL achieves competitive or better performance on 8 homophilic graph benchmarks and 6 heterophilic graph benchmarks, with significantly less computational overhead.

## 2 RELATED WORK

**Graph neural network on heterophilic graph.** Recently, the heterophily has been recognized as an important issue for graph neural networks, which is outlined by Pei et al. (2020) firstly. To make graph neural networks able to generalize well on the heterophilic graph, several efforts have been done from both the spatial and spectral perspectives (Pei et al., 2020; Abu-El-Haija et al., 2019; Zhu et al., 2020a; Chien et al., 2020; Li et al., 2021; Bo et al., 2021). Firstly, Chien et al. (2020) and Bo et al. (2021) analyze the necessary frequency component for GNNs to achieve good performance on heterophilic graphs and propose methods that are able to utilize high-frequency information. From the spatial perspective, several graph neural networks are designed to capture important dependencies between distant nodes (Pei et al., 2020; Abu-El-Haija et al., 2019; Bo et al., 2021; Zhu et al., 2020a). Although those methods have shown their effectiveness on heterophilic graphs, human annotations are required to guide the learning of neural networks.

**Graph contrastive learning.** Existing graph contrastive learning methods can be categorized into augmentation-based and augmentation-free methods, according to whether or not the graph augmentation techniques are employed during training. The augmentation-based methods (You et al., 2020; Peng et al., 2020; Hassani & Khasahmadi, 2020; Zhu et al., 2021a;b; 2020d; Thakoor et al., 2021; Zhu et al., 2020c) encourage the target graph encoder to be invariant to the manually specified graph transformations. Therefore, the design of graph augmentation is critical to the success of augmentation-based GCL. We summarized the augmentation methods commonly used by recent

works in Table 9 of Appendix D. Other works (Lee et al., 2021; Xia et al., 2022) try to get rid of the manual design of augmentation strategies, following the bootstrapped framework (Grill et al., 2020). They construct different views through two graph encoders updated with different strategies and push together the representations of the same node/class from different views. In both categories, those existing GCL methods require two graph forward-pass. Specifically, two augmented views of the same graph will be encoded separately by the same or two graph encoders for augmentation-based GCLs and the same graph will be encoded with two different graph encoders for augmentation-free GCLs, which is prohibitively expensive for large graphs. Besides, the theoretical analysis for the performance of GCL in the downstream tasks is still lacking. Although several efforts have been made in the visual domain (Arora et al., 2019; Lee et al., 2020; Tosh et al., 2021; HaoChen et al., 2021), the analysis for image classification cannot be trivially extended to graph setting, since the non-Euclidean graph structure is far more complex.

## 3 PRELIMINARY

**Notation.** Let $\mathcal{G} = (\mathcal{V}, \mathcal{E})$ denote an undirected graph, where $\mathcal{V} = \{v_i\}_{i \in [N]}$ and $\mathcal{E} \subseteq \mathcal{V} \times \mathcal{V}$ denote the node set and the edge set respectively. We denote the number of nodes and edges as $N$ and $E$, and the label of nodes as $\mathbf{y} \in \mathbb{R}^N$, in which $y_i \in [1, c], c \geq 2$ is the number of classes. The associated node feature matrix denotes as $\mathbf{X} \in \mathbb{R}^{N \times F}$, where $\mathbf{x}_i \in \mathbb{R}^F$ is the feature of node $v_i \in \mathcal{V}$ and $F$ is the input feature dimension. We denote the adjacent matrix as $\mathbf{A} \in \{0, 1\}^{N \times N}$, where $\mathbf{A}_{ij} = 1$ if $(v_i, v_j) \in \mathcal{E}$; and the corresponding degree matrix as $\mathbf{D} = \mathrm{diag}(d_1, \ldots, d_N)$, $d_i = \sum_j \mathbf{A}_{i,j}$. Our objective is to unsupervisedly learn a GNN encoder $f_\theta : \mathbf{X}, \mathbf{A} \to \mathbb{R}^{N \times K}$ receiving the node features and graph structure as input, that produces node representations in low dimensionality, i.e., $K \ll F$. The representations can benefit the downstream supervised or semi-supervised tasks, e.g., node classification.

**Homophilic and heterophilic graph.** Various metrics have been proposed to measure the homophily degree of a graph. Here we adopt two representative metrics, namely, node homophily and edge homophily. The edge homophily (Zhu et al., 2020b) is the proportion of edges that connect two nodes of the same class: $h_{edge} = \frac{|\{(v_i, v_j):(v_i, v_j) \in \mathcal{E} \wedge y_i = y_j\}|}{E}$, And the node homophily (Pei et al., 2020) is defined as, $h_{node} = \frac{1}{N} \sum_{v_i \in \mathcal{V}} \frac{|\{v_j:(v_i, v_j) \in \mathcal{E} \wedge y_i = y_j\}|}{|\{v_j:(v_i, v_j) \in \mathcal{E}\}|}$, which evaluates the average proportion of edge-label consistency of all nodes. They are all in the range of $[0, 1]$ and a value close to 1 corresponds to strong homophily while a value close to 0 indicates strong heterophily. As conventional, we refer the graph with high homophily degree as homophilic graph, and the graph with a low homophily degree as heterophilic graph. And we provided the homophily degree of graph considered in this work in Table 7 of Appendix A.1.

## 4 THEORETICAL ANALYSES

In this section, we firstly show the property of node representations obtained through the neighbor aggregation (Lemma 1). Then, based on the property, we introduce the single-pass graph contrastive loss (Equation (5)), in which the contrastive pairs are constructed according to the node similarity, instead of the graph augmentations. And Theorem 1 shows the viability of the pair selection through the node similarity computed based on node feature. We bridge the graph contrastive loss and the Matrix Factorization (Lemma 2). Then, leveraging the analysis for matrix factorization, we obtain the performance guarantee for the embedding learned with SP-GCL in the downstream node classification task (Theorem 2).

### 4.1 ANALYSIS OF AGGREGATED FEATURES

**Assumptions on graph data.** To obtain analytic and conceptual insights of the aggregated features, we firstly describe the graph data we considered. We assume that the node feature follows the Gaussian mixture model (Reynolds, 2009). For simplicity, we focus on the binary classification problem. Conditional on the (binary-) label $y$ and a latent vector $\boldsymbol{\mu} \sim \mathcal{N}(\mathbf{0}, \mathbf{I}_F/F)$ where the identity matrix $\mathbf{I}_F \in \mathbb{R}^{F \times F}$, the features are governed by:

$$\mathbf{x}_i = y_i \boldsymbol{\mu} + \frac{\mathbf{q}_i}{\sqrt{F}}, \tag{1}$$

where random variable $\mathbf{q}_i \in \mathbb{R}^F$ has independent standard normal entries and $y_i \in \{-1, 1\}$ representing latent classes with abuse of notation. Then, the features of nodes with class $y_i$ follow

the same distribution depending on $y_i$, *i.e.*, $\mathbf{x}_i \sim P_{y_i}(\mathbf{x})$. Furthermore, we make an assumption about the neighborhood patterns, For node $i$, its neighbor's labels are independently sampled from a distribution $P(y_i)$.

**Remark.** The above assumption implies that the feature of a node depends on its label and the neighbor's label is generated from distribution only dependent on the label of the central node, which contains both cases of *homophily and heterophily*.

With this assumption, we present the following Lemma 1, where we denote the learned embedding through the neighbor aggregation and a linear projection by $\mathbf{Z}$, and $\mathbf{Z}_i$ is the learned embedding with respect to input $\mathbf{x}_i$, and the $\mathbf{W}$ denotes the linear weight.

**Lemma 1 (Concentration Property of Aggregated Features)** *Consider a graph $G$ following the graph data assumption and Eq. (1), then the expectation of embedding is given by*

$$\mathbb{E}[\mathbf{Z}_i] = \mathbf{W} \, \mathbb{E}_{y \sim P(y_i), \mathbf{x} \sim P_y(\mathbf{x})}[\mathbf{x}], \tag{2}$$

*Furthermore, with a probability at least $1 - \delta$ over the distribution for the graph, we have:*

$$\|\mathbf{Z}_i - \mathbb{E}[\mathbf{Z}_i]\|_2 \leq \sqrt{\frac{\sigma_{\max}^2(\mathbf{W}) F \log(2F/\delta)}{2 \mathbf{D}_{ii} \|\mathbf{x}\|_{\psi_2}}} \tag{3}$$

*and*

$$\left| \mathbf{Z}_i^\top \mathbf{Z}_j - \mathbb{E}[\mathbf{Z}_i^\top \mathbf{Z}_j] \right| \leq \sqrt{\frac{\sigma_{\max}^2(\mathbf{W}^\top \mathbf{W}) \log(2N^2/\delta)}{2D^2 \|\mathbf{x}^2\|_{\psi_1}}} \tag{4}$$

*where $\mathbf{D}_{ii}$ is the degree of node $v_i$, $D \equiv \min_i \mathbf{D}_{ii}$, and the sub-gaussian norms $\|\mathbf{x}\|_{\psi_2} \equiv \min_i \|\mathbf{x}_{i,d}\|_{\psi_2}$, sub-exponential norms $\|\mathbf{x}^2\|_{\psi_1} \equiv \min_i \|\mathbf{x}_{i,d}^2\|_{\psi_1}$ for $d \in [1, F]$. Besides, $\sigma_{\max}^2(\mathbf{W})$ is the largest singular value of $\mathbf{W}$.*

**Remark.** Extending from Theorem 1 in (Ma et al., 2021), the above lemma indicates that, for any graph following the graph data assumption, *(i)* in expectation, nodes with the same label have the same embedding, Equation (2); *(ii)* the embeddings of nodes with the same label tends to concentrate onto a certain area in the embedding space, which can be regarded as the inductive bias of the neighborhood aggregation mechanism with the graph data assumption, Equation (3); *(iii)* the inner product of hidden representations approximates to its expectation with a high probability, Equation (4); *(iv)* with commonly used initialization, e.g. Kaiming initialization (He et al., 2015) and Lecun initialization (LeCun et al., 2012), the $\sigma_{\max}^2(\mathbf{W}^\top \mathbf{W})$ is bounded and the concentration is relatively tight. (Proof and detailed discussion are in Appendix C.1 and C.5 respectively.)

Although the Gaussian mixture modeling on the feature, the neighborhood patterns modeling on the graph structure and the linearization on the graph neural network are commonly adopted in several recent works for the theoretical analysis of GNNs (Deshpande et al., 2018; Baranwal et al., 2021; Ma et al., 2021), and their high-level conclusions still hold empirically for a wide range of real-world cases, we empirically verify the derived Concentration Property (Lemma 1) on the multi-class real-world graph data and the non-linear graph neural network ( Section 6.3 and Table 6) and provide a discussion about the empirical observations of the concentration property in other recent works (Wang et al., 2022b; Trivedi et al., 2022) (Appendix E.1).

## 4.2 SINGLE-PASS GRAPH CONTRASTIVE LOSS

In order to learn a more compact and linearly separable embedding space, we introduce the single-pass graph contrastive loss based on the property of aggregated features. Exploiting the concentration property explicitly, we regard nodes with small distance in the embedding space as positive pairs, and nodes with large distance as negative pairs. We formally define the positive and negative pairs to introduce the loss. We draw a node $v_i$ uniformly from the node set $\mathcal{V}$, $v_i \sim Uni(\mathcal{V})$, and draw the node $v_{i+}$ uniformly from the set $S^i$, where the set $S^i$ is consisted by the $K_{pos}$ nodes closest to node $v_i$. Concretely, $S^i = \{v_i^1, v_i^2, \ldots, v_i^{K_{pos}}\} = \arg\max_{v_j \in \mathcal{V}} (\mathbf{Z}_i^\top \mathbf{Z}_j, K_{pos})$, where $K_{pos} \in \mathbb{Z}^+$ and $\arg\max(\cdot, K_{pos})$ denotes the operator for the top-$K_{pos}$ selection. The two sampled nodes form a positive pair $(v_i, v_{i+})$. Two nodes $v_j$ and $v_k$, sampled independently from the node set, can be regarded as a negative pair $(v_j, v_k)$ (Arora et al., 2019). Following the insight of Contrastive

Learning (Wang & Isola, 2020), similar sample pairs stay close to each other while dissimilar ones are far apart, the Single-Pass Graph Contrastive Loss is defined as,

$$\mathcal{L}_{\text{SP-GCL}} = -2 \mathbb{E}_{\substack{v_i \sim Uni(\mathcal{V}) \\ v_{i+} \sim Uni(S_{pos}^i)}} \left[ \mathbf{Z}_i^\top \mathbf{Z}_{i+} \right] + \mathbb{E}_{\substack{v_j \sim Uni(\mathcal{V}) \\ v_k \sim Uni(\mathcal{V})}} \left[ \left( \mathbf{Z}_j^\top \mathbf{Z}_k \right)^2 \right]. \tag{5}$$

### 4.3 PERFORMANCE GUARANTEE FOR LEARNING LINEAR CLASSIFIER

For the convenience of analysis, we firstly introduce the concept of *transformed graph* as follows, which is constructed based on the original graph and the selected positive pairs.

**Definition 1 (Transformed Graph)** *Given the original graph $\mathcal{G}$ and its node set $\mathcal{V}$, the transformed graph, $\widehat{\mathcal{G}}$, has the same node set $\mathcal{V}$ but with the selected positive pairs as the edge set, $\widehat{\mathcal{E}} = \bigcup_i \{(v_i, v_i^k)|_{k=1}^K\}$.*

Note, the transformed graph is formed by positive pairs selected based on aggregated features. Coupling with the *Concentration Property of Aggregated Features* (Lemma 1), the transformed graph tends to have a larger homophily degree than the original graph. We provide more empirical verifications in Section 6.3.

The transformed graph is illustrated in Figure 1. We denote the adjacency matrix of transformed graph as $\widehat{\mathbf{A}} \in \{0,1\}^{N \times N}$, the number of edges as $\hat{E}$, and the symmetric normalized matrix as $\widehat{\mathbf{A}}_{sym} = \widehat{\mathbf{D}}^{-1/2} \widehat{\mathbf{A}} \widehat{\mathbf{D}}^{-1/2}$, where $\widehat{\mathbf{D}} = \text{diag}\left(\hat{d}_1, \ldots, \hat{d}_N\right), \hat{d}_i = \sum_j \widehat{\mathbf{A}}_{i,j}$. Correspondingly, we denote the symmetric normalized Laplacian as $\widehat{\mathbf{L}}_{sym} = \mathbf{I} - \widehat{\mathbf{A}}_{sym} =$

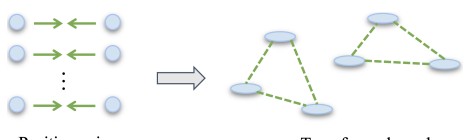

Positive pairs             Transformed graph

**Figure 1:** Transformed Graph formed with Positive Pairs.

$\mathbf{U}\mathbf{\Lambda}\mathbf{U}^\top$. Here $\mathbf{U} \in \mathbb{R}^{N \times N} = [\mathbf{u}_1, \ldots, \mathbf{u}_N]$, where $\mathbf{u}_i \in \mathbb{R}^N$ denotes the $i$-th eigenvector of $\widehat{\mathbf{L}}_{sym}$ and $\mathbf{\Lambda} = \text{diag}(\lambda_1, \ldots, \lambda_N)$ is the corresponding eigenvalue matrix. $\lambda_1$ and $\lambda_N$ be the smallest and largest eigenvalue respectively.

Then we show that optimizing a model with the contrastive loss (Equation (5)) is equivalent to the matrix factorization over the transformed graph, as stated in the following lemma:

**Lemma 2** *Denote the learnable embedding for matrix factorization as $\mathbf{F} \in \mathbb{R}^{N \times K}$. Let $\mathbf{F}_i = F_\psi(v_i)$. Then, the matrix factorization loss function $\mathcal{L}_{\text{mf}}$ is equivalent to the contrastive loss, Equation (5), up to an additive constant:*

$$\mathcal{L}_{\text{mf}}(\mathbf{F}) = \left\| \widehat{\mathbf{A}}_{sym} - \mathbf{F}\mathbf{F}^\top \right\|_F^2 = \mathcal{L}_{\text{SP-GCL}} + const \tag{6}$$

The above lemma bridges the graph contrastive learning and the graph matrix factorization and therefore allows us to provide the performance guarantee of SP-GCL by leveraging the power of matrix factorization. We leave the derivation in Appendix C.2. With the Lemma 1 and 2, we arrive at a conclusion about the expected value of the inner product of embeddings (More details in Appendix C.3):

**Theorem 1** *For a graph $G$ following the graph data assumption, then when the optimal of the contrastive loss is achieved, i.e., $\mathcal{L}_{\text{mf}}(\mathbf{F}^*) = 0$, we have,*

$$\mathbb{E}[\mathbf{Z}_i^\top \mathbf{Z}_j|_{y_i = y_j}] - \mathbb{E}[\mathbf{Z}_i^\top \mathbf{Z}_j|_{y_i \neq y_j}] \geq 1 - \bar{\phi}, \tag{7}$$

*where $\bar{\phi} = \mathbb{E}_{v_i, v_j \sim Uni(\mathcal{V})}\left(\widehat{\mathbf{A}}_{i,j} \cdot \mathbb{1}[y_i \neq y_j]\right)$.*

**Remark.** The theorem shows that the embedding inner product of nodes from the same class is larger than the inner product of nodes from different classes. Besides, the $1 - \bar{\phi}$, indicating the probability of an edge connecting two nodes from the same class, can be regarded as the edge homophily of the transformed graph. Therefore, the theorem implies that if the edge homophily of the transformed graph is larger, embeddings of nodes from the same class will be more compact in the high dimensional space.

Finally, with the Theorem 1 and Lemma 1, we obtain a performance guarantee for node embeddings learned by SP-GCL with a linear classifier in the downstream task (More details in Appendix C.4):

**Theorem 2** *Let $f^*_{\text{SP-GCL}} \in \arg\min_{f:\mathcal{X}\to\mathbb{R}^K}$ be a minimizer of the contrastive loss, $\mathcal{L}_{\text{SP-GCL}}$. Then there exists a linear classifier $\mathbf{B}^* \in \mathbb{R}^{c\times K}$ with norm $\|\mathbf{B}^*\|_F \leq 1/(1-\lambda_K)$ such that, with a probability at least $1-\delta$,*

$$\mathbb{E}_{v_i}\left[\left\|\vec{y}_i - \mathbf{B}^* f^*_{\text{gcl}}(v)\right\|_2^2\right] \leq \frac{\bar{\phi}}{\hat{\lambda}_{K+1}} + \sqrt{\frac{\sigma_{\max}^2(\mathbf{W}^\top\mathbf{W})\log(2N^2/\delta)}{2D^2\|\mathbf{x}^2\|_{\psi_1}\hat{\lambda}_{K+1}^2}}, \tag{8}$$

*$\hat{\lambda}_i$ are the $i$ smallest eigenvalues of the symmetrically normalized Laplacian matrix of the transformed graph.*

## 5 SINGLE-PASS GRAPH CONTRASTIVE LEARNING (SP-GCL)

As a direct consequence of our theory, we introduce the Single-Pass Graph Contrastive Learning (SP-GCL) to verify our findings. Instead of relying on the graph augmentation function or the exponential moving average, our new learning framework only forwards single time and the contrastive pairs are constructed based on the aggregated features. As we shall see, this exceedingly simple, theory motivated method also yields better performance in practice compared to dual-pass graph contrastive learning methods on both homophilic and heterophilic graph benchmarks. As the analysis revealed, for each class, the embedding obtained from neighbor aggregation will concentrate toward the expectation of embedding belonging to the class. Inspired by this, we design the self-supervision signal based on the obtained embedding and propose a novel single-pass graph contrastive learning framework, SP-GCL, which selects similar nodes as positive node pairs. As

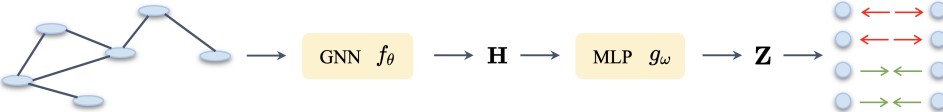

**Figure 2:** Overview of SP-GCL. The graph data is encoded by a graph neural network $f_\theta$ and a following projection head $g_\omega$. The contrastive pairs are constructed based on the representation $\mathbf{Z}$.

shown in Figure 2, In each iteration, the proposed framework firstly encodes the graph with a graph encoder $f_\theta$ denoted by $\mathbf{H} = f_\theta(\mathbf{X},\mathbf{A})$. Then, a projection head with *L2*-normalization, $g_\omega$, is employed to project the node embedding into the hidden representation $\mathbf{Z} = g_\omega(\mathbf{H})$. To scale up SP-GCL on large graphs, the the node pool, $P$, are constructed by the $T$-hop neighborhood of $b$ nodes (the seed node set $S$) uniformly sampled from $\mathcal{V}$. For each seed node $v_i \in S$, the top-$K_{pos}$ nodes with highest similarity from the node pool are selected as positive set for it which denote as $S^i_{pos} = \arg\max_{v_j \in P}(\mathbf{Z}_i^\top\mathbf{Z}_j, K_{pos})$, and $K_{neg}$ nodes are sampled from $\mathcal{V}$ to form the negative set $S^i_{neg}$, $S^i_{neg} \subseteq \mathcal{V}$. Concretely, the framework is optimized with the following objective:

$$\widehat{\mathcal{L}}_{\text{SP-GCL}} = -\frac{2}{N\cdot K_{pos}}\sum_{v_i\in\mathcal{V}}\sum_{v_{i+}\in S^i_{pos}}\left[\mathbf{Z}_i^\top\mathbf{Z}_{i+}\right] + \frac{1}{N\cdot K_{neg}}\sum_{v_j\in\mathcal{V}}\sum_{v_k\in S^i_{neg}}\left[\left(\mathbf{Z}_j^\top\mathbf{Z}_k\right)^2\right], \tag{9}$$

Notably, the empirical contrastive loss is an unbiased estimation of the Equation (5). Overall, the training algorithm SP-GCL is summarized in Algorithm 1.

Although we provide a theoretical discussion about the manner of self-selection for positive pairs and the proposed method, whether the method is effective and whether the self-selected manner is feasible for real-world cases are still not answered. In the following section, we empirically verify the effectiveness of the method and usefulness of our findings over a wide range of graph datasets.

## 6 EXPERIMENTS

### 6.1 PERFORMANCE ON HOMOPHILIC AND HETEROPHILIC GRAPH

The homophilic graph benchmarks have been studied by several previous works (Velickovic et al., 2019; Peng et al., 2020; Hassani & Khasahmadi, 2020; Zhu et al., 2020d; Thakoor et al., 2021; Lee et al., 2021). We re-use their configuration and compare SP-GCL with those methods and leave the detailed description about the experiment setting in Appendix A. And we leave the implementation details and the hyperparameter selection in Appendix A.4. The result is summarized in Table 1, in which the best performance achieved by self-supervised methods is marked in boldface.

**Algorithm 1:** Single-Pass Graph Contrastive Learning (SP-GCL).

---

**Input:** Graph neural network $f_\theta$, MLP projection head $g_\omega$, input adjacency matrix $\mathbf{A}$, node features $\mathbf{X}$, batch size $b$, number of hops $T$, number of positive nodes $K_{pos}$.

**for** epoch $\leftarrow 1, 2, \cdots$ **do**

    1. Obtain the node embedding, $\mathbf{H} = f_\theta(\mathbf{X}, \mathbf{A})$.

    2. Obtain the hidden representation, $\mathbf{Z} = g_\omega(\mathbf{H})$.

    3. Sample $b$ nodes as seed node set $S$ and construct the node pool $P$ with the $T$-hop neighbors of each node in the set $S$.

    4. Select top-$K_{pos}$ similar nodes for every $v_i \in S$ to form the positive node set $S_{pos}^i$.

    5. Compute the contrastive objective with Eq. (5) and update parameters by applying stochastic gradient.

**end for**

**return** Final model $f_\theta$.

---

Compared with augmentation-based and augmentation-free GCL methods, SP-GCL outperforms previous methods on 2 datasets and achieves competitive performance on the others, which shows the effectiveness of the single-pass contrastive loss on homophilic graphs. We further assess the model performance on heterophilic graph benchmarks that employed in Pei et al. (Pei et al., 2020) and Lim et al. (Lim et al., 2021a). As shown in Table 2, SP-GCL achieves the best performance on 6 of 6 heterophilic graphs by an evident margin. The above result indicates that, instead of relying on the augmentations which are sensitive to the graph type, **SP-GCL** *is able to work well over a wide range of real-world graphs (described in Appendix A.1) with different homophily degree*.

**Table 1:** Graph Contrastive Learning on Homophilic Graphs. The highest performance of unsupervised models is highlighted in boldface. OOM indicates Out-Of-Memory on a 32GB GPU.

| Model | Cora | CiteSeer | PubMed | WikiCS | Amz-Comp. | Amz-Photo | Coauthor-CS | Coauthor-Phy. |
|---|---|---|---|---|---|---|---|---|
| MLP | $47.92 \pm 0.41$ | $49.31 \pm 0.26$ | $69.14 \pm 0.34$ | $71.98 \pm 0.42$ | $73.81 \pm 0.21$ | $78.53 \pm 0.32$ | $90.37 \pm 0.19$ | $93.58 \pm 0.41$ |
| GCN | $81.54 \pm 0.68$ | $70.73 \pm 0.65$ | $79.16 \pm 0.25$ | $93.02 \pm 0.11$ | $86.51 \pm 0.54$ | $92.42 \pm 0.22$ | $93.03 \pm 0.31$ | $95.65 \pm 0.16$ |
| DeepWalk | $70.72 \pm 0.63$ | $51.39 \pm 0.41$ | $73.27 \pm 0.86$ | $74.42 \pm 0.13$ | $85.68 \pm 0.07$ | $89.40 \pm 0.11$ | $84.61 \pm 0.22$ | $91.77 \pm 0.15$ |
| Node2cec | $71.08 \pm 0.91$ | $47.34 \pm 0.84$ | $66.23 \pm 0.95$ | $71.76 \pm 0.14$ | $84.41 \pm 0.14$ | $89.68 \pm 0.19$ | $85.16 \pm 0.04$ | $91.23 \pm 0.07$ |
| GAE | $71.49 \pm 0.41$ | $65.83 \pm 0.40$ | $72.23 \pm 0.71$ | $73.97 \pm 0.16$ | $85.27 \pm 0.19$ | $91.62 \pm 0.13$ | $90.01 \pm 0.71$ | $94.92 \pm 0.08$ |
| VGAE | $77.31 \pm 1.02$ | $67.41 \pm 0.24$ | $75.85 \pm 0.62$ | $75.56 \pm 0.28$ | $86.40 \pm 0.22$ | $92.16 \pm 0.12$ | $92.13 \pm 0.16$ | $94.46 \pm 0.13$ |
| DGI | $82.34 \pm 0.71$ | $71.83 \pm 0.54$ | $76.78 \pm 0.31$ | $75.37 \pm 0.13$ | $84.01 \pm 0.52$ | $91.62 \pm 0.42$ | $92.16 \pm 0.62$ | $94.52 \pm 0.47$ |
| GMI | $82.39 \pm 0.65$ | $71.72 \pm 0.15$ | $79.34 \pm 1.04$ | $74.87 \pm 0.13$ | $82.18 \pm 0.27$ | $90.68 \pm 0.18$ | OOM | OOM |
| MVGRL | $\mathbf{83.45 \pm 0.68}$ | $\mathbf{73.28 \pm 0.48}$ | $80.09 \pm 0.62$ | $77.51 \pm 0.06$ | $87.53 \pm 0.12$ | $91.74 \pm 0.08$ | $92.11 \pm 0.14$ | $95.33 \pm 0.05$ |
| GRACE | $81.92 \pm 0.89$ | $71.21 \pm 0.64$ | $\mathbf{80.54 \pm 0.36}$ | $78.19 \pm 0.10$ | $86.35 \pm 0.44$ | $92.15 \pm 0.25$ | $92.91 \pm 0.20$ | $95.26 \pm 0.22$ |
| GCA | $82.07 \pm 0.10$ | $71.33 \pm 0.37$ | $80.21 \pm 0.39$ | $78.40 \pm 0.13$ | $87.85 \pm 0.31$ | $92.49 \pm 0.11$ | $92.87 \pm 0.14$ | $95.68 \pm 0.05$ |
| BGRL | $81.44 \pm 0.72$ | $71.82 \pm 0.48$ | $80.18 \pm 0.63$ | $76.96 \pm 0.61$ | $89.62 \pm 0.37$ | $93.07 \pm 0.34$ | $92.67 \pm 0.21$ | $95.47 \pm 0.28$ |
| AFGRL | $81.60 \pm 0.54$ | $71.02 \pm 0.37$ | $80.02 \pm 0.48$ | $77.98 \pm 0.49$ | $89.66 \pm 0.40$ | $\mathbf{93.14 \pm 0.36}$ | $\mathbf{93.27 \pm 0.17}$ | $\mathbf{95.69 \pm 0.10}$ |
| SP-GCL | $83.16 \pm 0.13$ | $71.96 \pm 0.42$ | $79.16 \pm 0.73$ | $\mathbf{79.01 \pm 0.51}$ | $\mathbf{89.68 \pm 0.19}$ | $92.49 \pm 0.31$ | $91.92 \pm 0.10$ | $95.12 \pm 0.15$ |

**Table 2:** Graph Contrastive Learning on Heterophilic Graphs. The highest performance of unsupervised models is highlighted in boldface. OOM indicates Out-Of-Memory on a 32GB GPU.

| Model | Chameleon | Squirrel | Actor | Twitch-DE | Twitch-gamers | Genius |
|---|---|---|---|---|---|---|
| MLP | $47.59 \pm 0.73$ | $31.67 \pm 0.61$ | $35.93 \pm 0.61$ | $69.44 \pm 0.67$ | $60.71 \pm 0.18$ | $86.62 \pm 0.11$ |
| GCN | $66.45 \pm 0.48$ | $53.03 \pm 0.57$ | $28.79 \pm 0.23$ | $73.43 \pm 0.71$ | $62.74 \pm 0.03$ | $87.72 \pm 0.18$ |
| DeepWalk | $43.99 \pm 0.67$ | $30.90 \pm 1.09$ | $25.50 \pm 0.28$ | $70.39 \pm 0.77$ | $61.71 \pm 0.41$ | $68.98 \pm 0.15$ |
| Node2cec | $31.49 \pm 1.17$ | $27.64 \pm 1.36$ | $27.04 \pm 0.56$ | $70.70 \pm 1.15$ | $61.12 \pm 0.29$ | $67.96 \pm 0.17$ |
| GAE | $39.13 \pm 1.34$ | $34.65 \pm 0.81$ | $25.36 \pm 0.23$ | $67.43 \pm 1.16$ | $56.26 \pm 0.50$ | $83.36 \pm 0.21$ |
| VGAE | $42.65 \pm 1.27$ | $35.11 \pm 0.92$ | $28.43 \pm 0.57$ | $68.62 \pm 1.82$ | $60.70 \pm 0.61$ | $85.17 \pm 0.52$ |
| DGI | $60.27 \pm 0.70$ | $42.22 \pm 0.63$ | $28.30 \pm 0.76$ | $72.77 \pm 1.30$ | $61.47 \pm 0.56$ | $86.96 \pm 0.44$ |
| GMI | $52.81 \pm 0.63$ | $35.25 \pm 1.21$ | $27.28 \pm 0.87$ | $71.21 \pm 1.27$ | OOM | OOM |
| MVGRL | $53.81 \pm 1.09$ | $38.75 \pm 1.32$ | $32.09 \pm 1.07$ | $71.86 \pm 1.21$ | OOM | OOM |
| GRACE | $61.24 \pm 0.53$ | $41.09 \pm 0.85$ | $28.27 \pm 0.43$ | $72.49 \pm 0.72$ | OOM | OOM |
| GCA | $60.94 \pm 0.81$ | $41.53 \pm 1.09$ | $28.89 \pm 0.50$ | $73.21 \pm 0.83$ | OOM | OOM |
| BGRL | $64.86 \pm 0.63$ | $46.24 \pm 0.70$ | $28.80 \pm 0.54$ | $73.31 \pm 1.11$ | $60.93 \pm 0.32$ | $86.78 \pm 0.71$ |
| AFGRL | $59.03 \pm 0.78$ | $42.36 \pm 0.40$ | $27.43 \pm 1.31$ | $69.11 \pm 0.72$ | OOM | OOM |
| SP-GCL | $\mathbf{65.28 \pm 0.53}$ | $\mathbf{52.10 \pm 0.67}$ | $\mathbf{28.94 \pm 0.69}$ | $\mathbf{73.51 \pm 0.97}$ | $\mathbf{62.04 \pm 0.17}$ | $\mathbf{90.06 \pm 0.18}$ |

**Table 3:** Computational requirements on a set of standard benchmark graphs. OOM indicates running out of memory on a 32GB GPU.

| Dataset | CS. | Phy. | Genius | Gamers. |
|---|---|---|---|---|
| # Nodes | 18,333 | 34,493 | 421,961 | 168,114 |
| # Edges | 327,576 | 991,848 | 984,979 | 6,797,557 |
| GRACE | 13.21 GB | 30.11 GB | OOM | OOM |
| BGRL | 3.10 GB | 5.42 GB | 8.18 GB | 26.22 GB |
| SP-GCL | 2.07 GB | 3.21 GB | 6.24 GB | 22.15 GB |

**Table 4:** The performance of SP-GCL with different hidden dimension. The average accuracy over 10 runs is reported.

| | WikiCS | Comp. | Actor | Twitch-DE. |
|---|---|---|---|---|
| $K = 256$ | 78.32 | 88.40 | 28.12 | 72.53 |
| $K = 512$ | 79.01 | 89.01 | 28.94 | 72.87 |
| $K = 1024$ | 79.07 | 89.68 | 29.87 | 73.51 |

## 6.2 COMPUTATIONAL COMPLEXITY ANALYSIS

In order to illustrate the advantages of SP-GCL, we provide a brief comparison of the time and space complexities between SP-GCL, the previous strong contrastive method, GCA (Zhu et al., 2021b), and the memory-efficient contrastive method, BGRL (Thakoor et al., 2021). Consider a graph with $N$ nodes and $E$ edges, and a graph neural network (GNN), $f$, that compute node embeddings in time and space $O(N + E)$. BGRL performs four GNN computations per update step, in which twice for the target and online encoders, and twice for each augmentation, and a node-level projection; GCA performs two GNN computations (once for each augmentation), plus a node-level projection. Both methods backpropagate the learning signal twice (once for each augmentation), and we assume the backward pass to be approximately as costly as a forward pass. Both of them will compute the augmented graphs by feature masking and edge masking on the fly, the cost for augmentation computation is nearly the same. Thus the total time and space complexity per update step for BGRL is $6C_{encoder}(E+N)+4C_{proj}N+C_{prod}N+C_{aug}$ and $4C_{encoder}(E+N)+4C_{proj}N+C_{prod}N^2+C_{aug}$ for GCA. The $C_{prod}$ depends on the dimension of node embedding and we assume the node embeddings of all the methods with the same size. For our proposed method, only one GNN encoder is employed and we compute the inner product of $b$ nodes to construct positive samples and $K_{pos}$ and $K_{neg}$ inner product for the loss computation. Then for SP-GCL, we have: $2C_{encoder}(E + N) + 2C_{proj}N + C_{prod}(K_{pos} + K_{neg})^2$. We empirically measure the peak of GPU memory usage of SP-GCL, GCA and BGRL. As a fair comparison, we set the embedding size as 128 for all those methods on the four datasets and keep the other hyper-parameters of the three methods the same as the main experiments. As shown by Table 3, *the computational overhead of* **SP-GCL** *is much less than the previous methods*.

**Table 5:** True positive ratio of selected edges at the end of training. The minimal, maximal, average, standard deviation of 20 runs are presented.

| Dataset | $h_{edge}$ of $\widehat{\mathcal{G}}$ |
|---|---|
| Cora | $0.812_{\uparrow} \pm 0.0022$ (0.810) |
| CiteSeer | $0.691_{\downarrow} \pm 0.0018$ (0.736) |
| PubMed | $0.819\uparrow \pm 0.0011$ (0.802) |
| Coauthor CS | $0.883\uparrow \pm 0.0018$ (0.808) |
| Coauthor Phy. | $0.952\uparrow \pm 0.0021$ (0.931) |
| Amazon Comp. | $0.866\uparrow \pm 0.0019$ (0.777) |
| Amazon Photo | $0.908\uparrow \pm 0.0012$ (0.827) |
| WikiCS | $0.751\uparrow \pm 0.0027$ (0.654) |
| Chameleon | $0.631\uparrow \pm 0.0047$ (0.234) |
| Squirrel | $0.526\uparrow \pm 0.0042$ (0.223) |
| Actor | $0.378\uparrow \pm 0.0026$ (0.216) |
| Twitch-DE | $0.669\uparrow \pm 0.0033$ (0.632) |
| Twitch-gamers | $0.617\uparrow \pm 0.0021$ (0.545) |
| Genius | $0.785\uparrow \pm 0.0028$ (0.618) |

**Table 6:** Edge homophily of the transformed graph at the initial stage. Homophily value of original graphs is shown in parentheses.

| Dataset | Min | Max | Avg | Std |
|---|---|---|---|---|
| Cora | 0.817 | 0.836 | 0.826 | 0.0061 |
| CiteSeer | 0.708 | 0.719 | 0.713 | 0.0033 |
| PubMed | 0.820 | 0.832 | 0.825 | 0.0037 |
| Coauthor CS | 0.892 | 0.905 | 0.901 | 0.0025 |
| Coauthor Phy. | 0.952 | 0.963 | 0.959 | 0.0028 |
| Amazon Comp. | 0.874 | 0.896 | 0.878 | 0.0076 |
| Amazon Photo | 0.903 | 0.926 | 0.916 | 0.0054 |
| WikiCS | 0.757 | 0.771 | 0.761 | 0.0040 |
| Chameleon | 0.703 | 0.785 | 0.711 | 0.0183 |
| Squirrel | 0.529 | 0.541 | 0.530 | 0.0024 |
| Actor | 0.388 | 0.399 | 0.396 | 0.0016 |
| Twitch-DE | 0.693 | 0.726 | 0.703 | 0.0073 |
| Twitch-gamers | 0.620 | 0.633 | 0.627 | 0.0022 |
| Genius | 0.786 | 0.797 | 0.790 | 0.0029 |

## 6.3 EMPIRICAL VERIFICATION FOR THE THEORETICAL ANALYSIS

**Effect of embedding dimension (*Scaling Behavior*).** We observe performance gains by scaling up models, as shown in Table 4. These observations are aligned with our analysis (Theorem 2), a larger hidden dimension, $K$, leads to better performance (larger $\hat{\lambda}_{K+1}$ leads to a lower bound). *Coupling with the efficiency property (Section 6.2), our scalable approach allows for learning high-capacity models that generalize well under the same computational requirement.*

**Homophily of transformed graph (*Initial Stage*).** The positive sampling depends largely on how the hidden representation is obtained. In other words, if starting from a "poor" initialization, the GNN encoder could yield false positive samples since the inner product is not correctly evaluated in

the beginning. Although, our analysis shows that the concentration property (Lemma 1) is relatively tight with commonly used initialization methods ($\sigma_{\max}(\mathbf{W}^\top \mathbf{W})$ of the Equation (4) is bounded, Appendix C.5) and indicates that the transformed graph formed by selected positive pairs will have large edge homophily degree, we empirically measure the edge homophily of the transformed graph at the beginning of the training stage with 20 runs. The mean and standard deviation are reported in Table 6. The edge homophily of all transformed graphs are larger than the original one with a small standard deviation, except the CiteSeer in which a relatively high edge homophily (0.691) can still be achieved. The result indicates that *the initialization is "good" enough at the beginning stage to form a useful transformed graph and, in turn, support the feasibility of the Lemma 1 over the real-world data.*

**Distance to class center and Node Coverage (*Learning Process*).** We measure change of the average cosine distance ($1 - CosineSimilarity$) between the node embeddings and the class-center embeddings during training. Specifically, the class-center embedding is the average of the node embeddings of the same class. As shown in Figure 3, we found that the node embeddings will concentrate on their corresponding class centers during training, which implies that *the learned embedding space becomes more compact and the learning process is stable. Intuitively, coupling with the "good" initialization as discussed above, the* SP-GCL *iteratively leverage the inductive bias in the stable learning process and refine the embedding space.* Furthermore, to study which nodes are benefited during the learning process, we measure the Node Cover Ratio and Overlapped Selection Ratio on four graph datasets. We denote the set of edges forming by the $K_{pos}$ positive selection at epoch $t$ as $e^t$ and define the Overlapped Selection Ratio as $\frac{e^{t+1} \cap e^t}{|e^{t+1}|}$. Besides, we denote the set of positive nodes at at epoch $t$ as $v^t$. Then the Node Cover Ratio at epoch $t$ is defined as: $\frac{|v^0 \cap v^1 \cdots v^t|}{N}$. Following the hyperparameters described in the previous section, we measure the Node Cover Ratio and Overlapped Selection Ratio during training. The results are shown in Figure 4, which shows that *all nodes are benefited from the optimization procedure.* Besides, we attribute the relatively small and non-increasing overlapped selection ratio to the batch training.

**Selected positive pairs (*End of Training*).** To further study the effect of the proposed method, we provide the true positive ratio (TPR) of the selected pairs at the end of training with 20 runs. As shown in Table 5, the relatively large TPR and low variance suggests that *the quality of the selected positive pairs is good and the learning process is stable.*

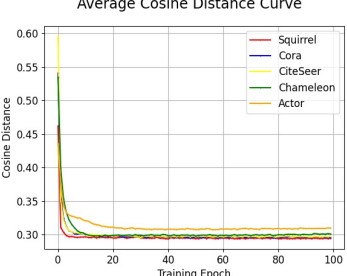

**Figure 3:** Average cosine distance between node embeddings and their corresponding class centers.

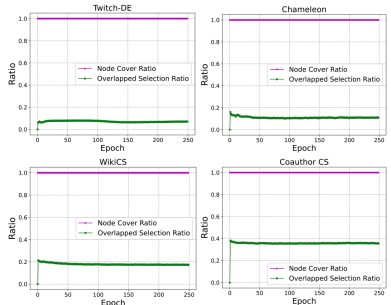

**Figure 4:** The Node Cover Ratio and Overlapped Selection Ratio.

## 7 CONCLUSION

In this work, we firstly analyze the concentration property of embedding obtained through neighborhood aggregation which holds for both homophilic and heterophilic graphs. Then, exploiting the concentration property, we introduce the single-pass graph contrastive loss. We theoretically show that the equivalence between the contrastive objective and the matrix factorization. Further, leveraging the analysis of the matrix factorization, we provide a theoretical guarantee for the node embedding, obtained through minimizing the contrastive loss, in the downstream classification task. To verify the usefulness of our findings in real-world datasets, we implement the Single-Pass Graph Contrastive Learning framework (SP-GCL). Empirically, we show that SP-GCL can outperform or be competitive with SOTA methods on 8 homophilic graph benchmarks and 6 heterophilic graph benchmarks with significantly less computational overhead. The empirical results verify the feasibility and effectiveness of our analysis in real-world cases. We leave the discussion about the connection with existing observation, limitation and future work in Appendix E.

## 8    REPRODUCIBILITY STATEMENT

To ensure the results and conclusions of our paper are reproducible, we make the following efforts:

Theoretically, we state the full set of assumptions and include complete proofs of our theoretical results in Section 4 and Appendix C.

Experimentally, we provide our code, and instructions needed to reproduce the main experimental results. And we specify all the training and implementation details in Section 6 and Appendix A. Besides, we independently run experiments and report the mean and standard deviation.

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

# A APPENDIX: EXPERIMENT SETTING

## A.1 DATASET INFORMATION

We analyze the quality of representations learned by SP-GCL on transductive node classification benchmarks. Specifically, we evaluate the performance of using the pretraining representations on 8 benchmark homophilic graph datasets, namely, Cora, Citeseer, Pubmed (Kipf & Welling, 2016a) and Wiki-CS, Amazon-Computers, Amazon-Photo, Coauthor-CS, Coauthor-Physics (Shchur et al., 2018), as well as 6 heterophilic graph datasets, namely, Chameleon, Squirrel (Rozemberczki et al., 2021), Actor (Pei et al., 2020), Twitch-DE, Twitch-gamers (Rozemberczki & Sarkar, 2021), and Genius (Lim et al., 2021b). The datasets are collected from real-world networks from different domains; their detailed statistics are summarized in Table 7. For the 8 homophilic graph data, we use the processed version provided by PyTorch Geometric (Fey & Lenssen, 2019). Besides, for the 6 heterophilic graph data, 3 of them, e.g., Chameleon, Squirrel and Actor are provided by PyTorch Geometric library. The other three dataset, genius, twitch-DE and twitch-gamers can be obtained from the official github repository[1], in which the standard splits for all the 6 heterophilic graph datasets can also be obtained. Those graph datasets follow the MIT license, and the personal identifiers are not included. We do not foresee any form of privacy issues.

**Table 7:** Statistics of datasets used in experiments.

| Name | Nodes | Edges | Classes | Feat. | $h_{node}$ | $h_{edge}$ |
|------|-------|-------|---------|-------|------------|------------|
| Cora | 2,708 | 5,429 | 7 | 1,433 | .825 | .810 |
| CiteSeer | 3,327 | 4,732 | 6 | 3,703 | .717 | .736 |
| PubMed | 19,717 | 44,338 | 3 | 500 | .792 | .802 |
| Coauthor CS | 18,333 | 327,576 | 15 | 6,805 | .832 | .808 |
| Coauthor Phy. | 34,493 | 991,848 | 5 | 8,451 | .915 | .931 |
| Amazon Comp. | 13,752 | 574,418 | 10 | 767 | .785 | .777 |
| Amazon Photo | 7,650 | 287,326 | 8 | 745 | .836 | .827 |
| WikiCS | 11,701 | 216,123 | 10 | 300 | .658 | .654 |
| Chameleon | 2,277 | 36,101 | 5 | 2,325 | .103 | .234 |
| Squirrel | 5,201 | 216,933 | 5 | 2,089 | .088 | .223 |
| Actor | 7,600 | 33,544 | 5 | 9,31 | .154 | .216 |
| Twitch-DE | 9,498 | 153,138 | 2 | 2,514 | .529 | .632 |
| Twitch-gamers | 168,114 | **6,797,557** | 2 | 7 | .552 | .545 |
| Genius | **421,961** | 984,979 | 2 | 12 | .477 | .618 |

## A.2 BASELINES

We consider representative baseline methods belonging to the following three categories (1) Traditional unsupervised graph embedding methods, including DeepWalk (Perozzi et al., 2014) and Node2Vec (Grover & Leskovec, 2016) , (2) Self-supervised learning algorithms with graph neural networks including Graph Autoencoders (GAE, VGAE) (Kipf & Welling, 2016b) , Deep Graph Infomax (DGI) (Velickovic et al., 2019) , Graphical Mutual Information Maximization (GMI) (Peng et al., 2020), and Multi-View Graph Representation Learning (MVGRL) (Hassani & Khasahmadi, 2020), graph contrastive representation learning (GRACE) (Zhu et al., 2020d) Graph Contrastive learning with Adaptive augmentation (GCA) (Zhu et al., 2021b), Bootstrapped Graph Latents (BGRL) (Thakoor et al., 2021), Augmentation-Free Graph Representation Learning (AFGRL) (Lee et al., 2021), (3) Supervised learning and Semi-supervised learning, e.g., Multilayer Perceptron (MLP) and Graph Convolutional Networks (GCN) (Kipf & Welling, 2016a), where they are trained in an end-to-end fashion.

## A.3 EVALUATION PROTOCOL

We follow the evaluation protocol of previous state-of-the-art graph contrastive learning approaches. Specifically, for every experiment, we employ the linear evaluation scheme as introduced in (Velickovic et al., 2019), where each model is firstly trained in an unsupervised manner; then, the pretrained

---

[1] https://github.com/CUAI/Non-Homophily-Large-Scale

representations are used to train and test via a simple linear classifier. For the datasets that came with standard train/valid/test splits, we evaluate the models on the public splits. For datasets without standard split, e.g., Amazon-Computers, Amazon-Photo, Coauthor-CS, Coauthor-Physics, we randomly split the datasets, where 10%/10%/80% of nodes are selected for the training, validation, and test set, respectively. For most datasets, we report the averaged test accuracy and standard deviation over 10 runs of classification. While, following the previous works (Lim et al., 2021b;a), we report the test ROC AUC on genius and Twitch-DE datasets.

## A.4 IMPLEMENTATION DETAILS

**Model architecture and hyperparamters.** We employ a two-layer GCN (Kipf & Welling, 2016a) as the encoder for all methods. The propagation for a single layer GCN is given by,

$$\text{GCN}_i(\mathbf{X}, \mathbf{A}) = \sigma\left(\bar{\mathbf{D}}^{-\frac{1}{2}}\bar{\mathbf{A}}\bar{\mathbf{D}}^{-\frac{1}{2}}\mathbf{X}\mathbf{W}_i\right),$$

where $\bar{\mathbf{A}} = \mathbf{A} + \mathbf{I}$ is the adjacency matrix with self-loops, $\bar{\mathbf{D}}$ is the degree matrix, $\sigma$ is a non-linear activation function, such as ReLU, and $\mathbf{W}_i$ is the learnable weight matrix for the $i$'th layer. Besides, following the previous works (Lim et al., 2021a;b), we use batch normalization within the graph encoder for the heterophilic graphs. The hyperparameters setting for all experiments are summarized in Table 8. We would like to release our code after acceptance.

**Linear evaluation of embeddings.** In the linear evaluation protocol, the final evaluation is over representations obtained from pretrained model. When fitting the linear classifier on top of the frozen learned embeddings, the gradient will not flow back to the encoder. We optimize the one layer linear classifier 1000 epochs using Adam with learning rate 0.0005.

**Hardware and software infrastructures.** Our model are implemented with PyTorch Geometric 2.0.3 (Fey & Lenssen, 2019), PyTorch 1.9.0 (Paszke et al., 2017). We conduct experiments on a computer server with four NVIDIA Tesla V100 SXM2 GPUs (with 32GB memory each) and twelve Intel Xeon Gold 6240R 2.40GHz CPUs.

**Table 8:** Hyperparameter settings for all experiments.

|  | $lr.$ | $K_{pos}$ | $K_{neg}$ | $b$ | $T$ | $K$ |
|---|---|---|---|---|---|---|
| Cora | .0010 | 5 | 100 | 512 | 2 | 1024 |
| CiteSeer | .0020 | 10 | 200 | 256 | 2 | 1024 |
| PubMed | .0010 | 5 | 100 | 256 | 2 | 1024 |
| WikiCS | .0010 | 5 | 100 | 512 | 2 | 1024 |
| Amz-Comp. | .0010 | 5 | 100 | 256 | 2 | 1024 |
| Amz-Photo | .0010 | 5 | 100 | 512 | 2 | 1024 |
| Coauthor-CS | .0010 | 5 | 100 | 512 | 2 | 1024 |
| Coauthor-Phy. | .0010 | 5 | 100 | 256 | 2 | 1024 |
| Chameleon | .0010 | 5 | 100 | 512 | 3 | 1024 |
| Squirrel | .0010 | 5 | 100 | 512 | 3 | 1024 |
| Actor | .0010 | 10 | 100 | 512 | 4 | 1024 |
| Twitch-DE | .0010 | 5 | 100 | 512 | 4 | 1024 |
| Twitch-gamers | .0010 | 5 | 100 | 256 | 4 | 128 |
| Genius | .0010 | 5 | 100 | 256 | 3 | 512 |

## B APPENDIX: ABLATION STUDY

### B.1 EFFECT OF TOP-K POSITIVE SAMPLING

We study the effect of top-$K$ positive sampling on the node classification performance by measuring the classification accuracy and the corresponding standard deviation with a wide range of $K$. The results over WikiCS, Chameleon, Squirrel, Cora, and Photo are summarized in Figure 5. We observe that the performance achieved by SP-GCL is insensitive to the selection of $K$ from 2 to 18.

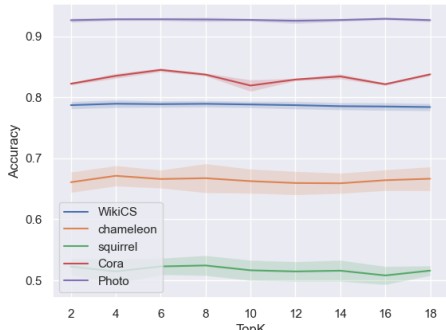 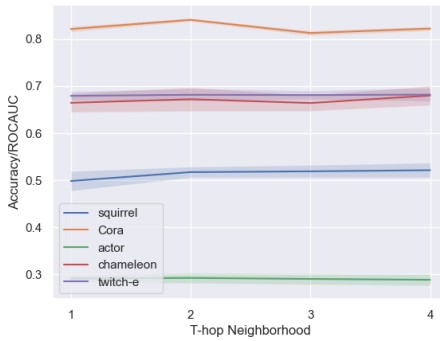

**Figure 5:** The effect of top-$K$ positive sampling on the Performance.

**Figure 6:** The effect of $T$-hop neighborhood on the Performance.

### B.2 EFFECT OF $T$-HOP NEIGHBORS

Factor $T$ is the other factor involved in node sampling. We provide a study about the the effect of $T$-hop neighbors on the node classification performance by measuring the classification accuracy or ROCAUC score and the corresponding standard deviation with different $T$. Specifically, with the same evaluation protocol described in Section A.3, we measure the classification accuracy on Squirrel, Cora, Actor, and Chameleon dataset and ROCAUC score on twitch-DE. As shown in Figure 6, the performance achieved by SP-GCL is insensitive to the selection of $T$.

## C APPENDIX: DETAILED PROOFS

### C.1 PROOF OF LEMMA 1

*Proof.* We first calculate the expectation of aggregated embedding:

$$\mathbb{E}[f_\theta(\mathbf{x}_i)] = \mathbb{E}\left[\mathbf{W} \sum_{j \in \mathcal{N}(i)} \frac{1}{\mathbf{D}_{ii}} \mathbf{x}_j\right] = \mathbf{W}\mathbb{E}_{y \sim P_{y_i}, \mathbf{x} \sim P_y(\mathbf{x})}[\mathbf{x}] \tag{10}$$

This equation is based on the graph data assumption such that $\mathbf{x}_j \sim P_{y_i}(\mathbf{x})$ for every $j$. Now we provide a concentration analysis. Because each feature $\mathbf{x}_i$ is a sub-Gaussian variable, then by Hoeffding's inequality, with probability at least $1 - \delta'$ for each $d \in [1, F]$, we have,

$$\left|\frac{1}{\mathbf{D}_{ii}} \sum_j (\mathbf{x}_{j,d} - \mathbb{E}[\mathbf{x}_{j,d}])\right| \leq \sqrt{\frac{\log(2/\delta')}{2\mathbf{D}_{ii}\|\mathbf{x}_{j,d}\|_{\psi_2}}} \tag{11}$$

where $\|\mathbf{x}_{j,d}\|_{\psi_2}$ is sub-Gaussian norm of $\mathbf{x}_{j,d}$. Furthermore, because each dimension of $\mathbf{x}_j$ is *i.i.d.*, thus we have $\|\mathbf{x}_j\|_{\psi_2} = \|\mathbf{x}_{j,d}\|_{\psi_2}$, for $d \in [1, F]$. Then we apply a union bound by setting $\delta' = F\delta$ on the feature dimension $k$. Then with probability at least $1 - \delta$ we have

$$\left|\frac{1}{\mathbf{D}_{ii}} \sum_j (\mathbf{x}_{j,d} - \mathbb{E}[\mathbf{x}_{j,d}])\right| \leq \sqrt{\frac{\log(2F/\delta)}{2\mathbf{D}_{ii}\|\mathbf{x}\|_{\psi_2}}} \tag{12}$$

Next, we use the matrix perturbation theory,

$$\left\|\frac{1}{\mathbf{D}_{ii}} \sum_j (\mathbf{x}_{j,d} - \mathbb{E}[\mathbf{x}_{j,d}])\right\|_2 \leq \sqrt{F}\left|\frac{1}{\mathbf{D}_{ii}} \sum_j (\mathbf{x}_{j,d} - \mathbb{E}[\mathbf{x}_{j,d}])\right|$$
$$\leq \sqrt{\frac{F\log(2F/\delta)}{2\mathbf{D}_{ii}\|\mathbf{x}\|_{\psi_2}}} \tag{13}$$

Finally, plug the weight matrix into the inequality,

$$\|f_\theta(\mathbf{x}_i) - \mathbb{E}[f_\theta(\mathbf{x}_i)]\| \leq \sigma_{\max}(\mathbf{W})\left\|\frac{1}{\mathbf{D}_{ii}} \sum_j (\mathbf{x}_{j,k} - \mathbb{E}[\mathbf{x}_{j,k}])\right\|_2 \tag{14}$$

where $\sigma_{\max}$ is the largest singular value of weight matrix.

Next, we perform a concentration analysis for the inner product. We first write down the detailed expression for each pair of $i, j$,

$$s_{i,j} \equiv \mathbf{x}_i^\top \mathbf{W}^\top \mathbf{W} \mathbf{x}_j \tag{15}$$

We first bound $\mathbf{x}_i^\top \mathbf{x}_j$. Because $\mathbf{x}_i$ and $\mathbf{x}_j$ are independently sampled from an identical distribution, then the product $\mathbf{x}_i^\top \mathbf{x}_j$ is sub-exponential. This can been seen from Orilicz norms relation that $\|\mathbf{x}^2\|_{\psi_1} = (\|\mathbf{x}^2\|_{\psi_2})^2$, where $\|\mathbf{x}\|_{\psi_2}$ is sub-exponential norm of $\mathbf{x}^2$. Then by the Hoeffding's inequality for sub-exponential variable, with a probability at least $1 - \delta$, we have

$$|\mathbf{x}_i^\top \mathbf{x}_j - \mathbb{E}_{\mathbf{x}_i \sim P_{y_i}, \mathbf{x}_j \sim P_{y_j}}[\mathbf{x}_i^\top \mathbf{x}_j]| \leq \sqrt{\frac{\sigma_{\max}^2(\mathbf{W}^\top \mathbf{W}) \log(2/\delta)}{2\|\mathbf{x}^2\|_{\psi_1}}} \tag{16}$$

Because that the aggregated feature is normalized by the degree of corresponding node, we have, for each pair of $i, j$

$$|s_{i,j} - \mathbb{E}[s_{i,j}]| \leq \sqrt{\frac{\log(2/\delta)\sigma_{\max}^2(\mathbf{W}^\top \mathbf{W})}{2\|\mathbf{x}^2\|_{\psi_1}\mathbf{D}_{ii}\mathbf{D}_{jj}}} \leq \sqrt{\frac{\sigma_{\max}^2(\mathbf{W}^\top \mathbf{W}) \log(2/\delta)}{2\|\mathbf{x}^2\|_{\psi_1} D^2}} \tag{17}$$

where $D = \min_i \mathbf{D}_{ii}$ for $i \in [1, N]$. Finally we apply a union bound over a pair of $i, j$. Then with probability at least $1 - \delta$ we have

$$|\mathbf{Z}_i^\top \mathbf{Z}_j - \mathbb{E}[\mathbf{Z}_i^\top \mathbf{Z}_j]| \leq \sqrt{\frac{\sigma_{\max}^2(\mathbf{W}^\top \mathbf{W}) \log(2N^2/\delta)}{2D^2 \|\mathbf{x}^2\|_{\psi_1}}} \tag{18}$$

### C.2 PROOF OF LEMMA 2

To prove this lemma, we first introduce the concept of the probability adjacency matrix. For the transformed graph $\widehat{\mathcal{G}}$, we denote its probability adjacency matrix as $\widehat{\mathbf{W}}$, in which $\hat{w}_{ij} = \frac{1}{\widehat{E}} \cdot \widehat{\mathbf{A}}_{ij}$. $\hat{w}_{ij}$ can be understood as the probability that two nodes have an edge and the weights sum to 1 because the total probability mass is 1: $\sum_{i,j} \hat{w}_{i,j'} = 1$, for $v_i, v_j \in \mathcal{V}$. The corresponding symmetric normalized matrix is $\widehat{\mathbf{W}}_{sym} = \widehat{\mathbf{D}}_{\mathbf{w}}^{-1/2} \widehat{\mathbf{W}} \widehat{\mathbf{D}}_{\mathbf{w}}^{-1/2}$, and the $\widehat{\mathbf{D}}_{\mathbf{w}} = diag([\hat{w}_1, \ldots, \hat{w}_N])$, where $\hat{w}_i = \sum_j \hat{w}_{ij}$. We then introduce the Matrix Factorization Loss which is defined as:

$$\min_{\mathbf{F} \in \mathbb{R}^{N \times k}} \mathcal{L}_{\mathrm{mf}}(\mathbf{F}) := \left\| \widehat{\mathbf{A}}_{sym} - \mathbf{F}\mathbf{F}^\top \right\|_F^2. \tag{19}$$

By the classical theory on low-rank approximation, Eckart-Young-Mirsky theorem (Eckart & Young, 1936), any minimizer $\widehat{\mathbf{F}}$ of $\mathcal{L}_{\mathrm{mf}}(\mathbf{F})$ contains scaling of the smallest eigenvectors of $\mathbf{L}_{sym}$ (also, the largest eigenvectors of $\widehat{\mathbf{A}}_{sym}$) up to a right transformation for some orthonormal matrix $\mathbf{R} \in \mathbb{R}^{K \times K}$. We have $\widehat{\mathbf{F}} = \mathbf{F}^*. \operatorname{diag}\left([\sqrt{1-\lambda_1}, \ldots, \sqrt{1-\lambda_K}]\right) \mathbf{R}$, where $\mathbf{F}^* = [\mathbf{u}_1, \mathbf{u}_2, \cdots, \mathbf{u}_K] \in \mathbb{R}^{N \times K}$. To proof the Lemma 2, we first present the Lemma 3.

**Lemma 3** *For transformed graph, its probability adjacency matrix $\widehat{\mathbf{W}}$, and adjacency matrix $\widehat{\mathbf{A}}$ are equal after the symmetric normalization, $\widehat{\mathbf{W}}_{sym} = \widehat{\mathbf{A}}_{sym}$.*
*Proof.* For any two nodes $v_i, v_j \in \mathcal{V}$ and $i \neq j$, we denote the the element in $i$-th row and $j$-th column of matrix $\widehat{\mathbf{W}}_{sym}$ as $\widehat{\mathbf{W}}_{sym}^{ij}$.

$$\widehat{\mathbf{W}}_{sym}^{ij} = \frac{1}{\sqrt{\sum_k \hat{w}_{ik}} \sqrt{\sum_k \hat{w}_{kj}}} \frac{1}{E} \widehat{\mathbf{A}}^{ij} = \frac{1}{\sqrt{\sum_k \widehat{\mathbf{A}}_{ik}} \sqrt{\sum_k \widehat{\mathbf{A}}_{kj}}} \widehat{\mathbf{A}}^{ij} = \widehat{\mathbf{A}}_{sym}^{ij}. \tag{20}$$

Leveraging the Lemma 3, we present the proof of Lemma 2.

**Proof of Lemma 2**  We start from the matrix factorization loss over $\widehat{\mathbf{A}}_{sym}$ to show the equivalence.

$$
\begin{aligned}
\|\widehat{\mathbf{A}}_{sym} - \mathbf{F}\mathbf{F}^\top\|_F^2 &= \|\widehat{\mathbf{W}}_{sym} - \mathbf{F}\mathbf{F}^\top\|_F^2 \\
&= \sum_{ij} \big( \frac{\hat{w}_{ij}}{\sqrt{\hat{w}_i}\sqrt{\hat{w}_j}} - f_{\mathrm{mf}}(v_i)^\top f_{\mathrm{mf}}(v_j) \big)^2 \\
&= \sum_{ij} (f_{\mathrm{mf}}(v_i)^\top f_{\mathrm{mf}}(v_j))^2 - 2\sum_{ij} \frac{\hat{w}_{ij}}{\sqrt{\hat{w}_i}\sqrt{\hat{w}_j}} f_{\mathrm{mf}}(v_i)^\top f_{\mathrm{mf}}(v_j) + \|\hat{\mathbf{W}}_{sym}\|_F^2 \\
&= \sum_{ij} \hat{w}_i \hat{w}_j \big[ \big( \frac{1}{\sqrt{\hat{w}_i}} \cdot f_{\mathrm{mf}}(v_i) \big)^\top \big( \frac{1}{\sqrt{\hat{w}_j}} \cdot f_{\mathrm{mf}}(v_j) \big) \big]^2 \\
&\quad - 2\sum_{ij} \hat{w}_{ij} \big( \frac{1}{\sqrt{\hat{w}_i}} \cdot f_{\mathrm{mf}}(v_i) \big)^\top \big( \frac{1}{\sqrt{\hat{w}_j}} \cdot f_{\mathrm{mf}}(v_j) \big) + C
\end{aligned}
\tag{21}
$$

where $f_{\mathrm{mf}}(v_i)$ is the $i$-th row of the embedding matrix $\mathbf{F}$. The $\hat{w}_i$ which can be understood as the node selection probability which is proportional to the node degree. Then, we can define the corresponding sampling distribution as $P_{deg}$. If and only if $\sqrt{w_i} \cdot F_\psi(v_i) = f_{\mathrm{mf}}(v_i) = \mathbf{F}_i$, the we have:

$$
\mathbb{E}_{\substack{v_i \sim P_{deg} \\ v_j \sim P_{deg}}} \big( F_\psi(v_i)^\top F_\psi(v_j) \big)^2 - 2\,\mathbb{E}_{\substack{v_i \sim Uni(\mathcal{V}) \\ v_{i+} \sim Uni(\mathcal{N}(v_i))}} \big( F_\psi(v_i)^\top F_\psi(v_{i+}) \big) + C
\tag{22}
$$

where $\mathcal{N}(v_i)$ denotes the neighbor set of node $v_i$ and $Uni(\cdot)$ is the uniform distribution over the given set. Because we constructed the transformed graph by selecting the top-$K_{pos}$ nodes for each node, then all nodes have the same degree. We can further simplify the objective as:

$$
\mathbb{E}_{\substack{v_i \sim Uni(\mathcal{V}) \\ v_j \sim Uni(\mathcal{V})}} \big( \mathbf{Z}_i^\top \mathbf{Z}_j \big)^2 - 2\,\mathbb{E}_{\substack{v_i \sim Uni(\mathcal{V}) \\ v_{i+} \sim Uni(S_{pos}^i)}} \big( \mathbf{Z}_i^\top \mathbf{Z}_{i+} \big) + C.
\tag{23}
$$

Due to the node selection procedure, the factor $\sqrt{w_i}$ is a constant and can be absorbed by the neural network, $F_\psi$. Then, because $\mathbf{Z}_i = F_\psi(v_i)$, we can have the Equation 23. Therefore, the minimizer of matrix factorization loss is equivalent with the minimizer of the contrastive loss.

### C.3  PROOF OF THEOREM 1

*Proof.* Now we provide the proof for the inner product of embedding. In particular, when we can achieve near-zero contrastive learning loss, *i.e.*, $\mathcal{L}_{\mathrm{mf}}(\mathbf{F}) = 0$, the minimizer $\mathbf{F}^*$ of $\mathcal{L}_{\mathrm{mf}}(\mathbf{F})$ contains scaling of the smallest eigenvectors of $\widehat{\mathbf{L}}_{sym}$ (also, the largest eigenvectors of $\widehat{\mathbf{A}}_{sym}$), $\mathbf{F}^* = [\mathbf{u}_1, \mathbf{u}_2, \cdots, \mathbf{u}_K] \in \mathbb{R}^{N \times K}$, according to the Eckart-Young-Mirsky theorem (Eckart & Young, 1936). Recall that $\mathbf{y} = \{1, -1\}^N \in \mathbb{R}^{N \times 1}$ is label of all nodes. Then we show there is a constraint on the quadratic form with respect to the optimal classifier. According to graph spectral theory (Chung & Graham, 1997), the quadratic form

$$
\mathbf{y}^\top \widehat{\mathbf{L}}_{sym} \mathbf{y} = \frac{1}{2} \sum_{i,i'} (\widehat{\mathbf{A}}_{sym})_{i,i'} (y_i - y_{i'})^2
$$

which captures the amount of edges connecting different labels. Furthermore, suppose that the expected homophily over distribution of graph feature and label, i.e., $y \sim P(y_i), \mathbf{x} \sim P_y(\mathbf{x})$, through similarity selection satisfies $\mathbb{E}[h_{edge}(\hat{\mathcal{G}})] = 1 - \bar{\phi}$. Here $\bar{\phi} = \mathbb{E}_{v_i, v_j \sim Uni(\mathcal{V})} \big( \widehat{\mathbf{A}}_{i,j} \cdot \mathbb{1}[y_i \neq y_j] \big)$. Since we have defined that $\bar{\phi}$ as the density of edges that connects different labels, we can directly show that $\mathbf{y}^\top \widehat{\mathbf{L}}_{sym} \mathbf{y} \leq \bar{\phi} N$. Then we expand the above expression accoding to $\widehat{\mathbf{L}}_{sym} = \mathbf{I} - \widehat{\mathbf{A}}_{sym}$,

$$
\mathbf{y}^\top \widehat{\mathbf{L}}_{sym} \mathbf{y} = \mathbf{y}^\top (\mathbf{I} - \widehat{\mathbf{A}}_{sym}) \mathbf{y} = N - \mathbf{y}^\top \widehat{\mathbf{A}}_{sym} \mathbf{y}
$$

Next we consider the situation that we achieve the optimal solution of the contrastive loss, $\mathbf{Z} = \arg\min \mathcal{L}_{\text{SP-GCL}}$. Then, we have $\mathcal{L}_{mf}(\mathbf{F}^*) = 0$, which implies that $\widehat{\mathbf{A}}_{sym} = (\mathbf{F}^*)^\top \mathbf{F}^*$. As we analyzed in Section C.2, we have $\mathbf{Z} = \mathbf{F}^*$ in this case. Furthermore, we have,

$$
\begin{aligned}
\mathbf{y}^\top \widehat{\mathbf{L}}_{sym} \mathbf{y} &= \mathbf{y}^\top (\mathbf{I} - \widehat{\mathbf{A}}_{sym}) \mathbf{y} = N - \mathbf{y}^\top \mathbf{Z}^\top \mathbf{Z} \mathbf{y} = N - \mathbf{y}^\top (\mathbf{Z}_i^\top \mathbf{Z}_j)_{(i,j \in n \times n)} \mathbf{y} \\
&= N - \big( \sum_{i,j} \mathbf{Z}_i^\top \mathbf{Z}_j |_{y_i = y_j} - \sum_{i,j} \mathbf{Z}_i^\top \mathbf{Z}_j |_{y_i \neq y_j} \big)
\end{aligned}
\tag{24}
$$

This leads to,

$$\frac{1}{N} \Big[ \sum_{i,j} \mathbf{Z}_i^\top \mathbf{Z}_j |_{y_i=y_j} - \sum_{i,j} \mathbf{Z}_i^\top \mathbf{Z}_j |_{y_i \neq y_j} \Big] = \mathbb{E}[\mathbf{Z}_i^\top \mathbf{Z}_j |_{y_i=y_j}] - \mathbb{E}[\mathbf{Z}_i^\top \mathbf{Z}_j |_{y_i \neq y_j}] \geq 1 - \bar{\phi} \tag{25}$$

## C.4 PROOF OF THEOREM 2

Recently, HaoChen et al. (2021) presented the following theoretical guarantee for the model learned with the matrix factorization loss.

**Lemma 4** *For a graph $\mathcal{G}$, let $f_{\mathrm{mf}}^* \in \arg\min_{f_{\mathrm{mf}}:\mathcal{V}\to\mathbb{R}^K}$ be a minimizer of the matrix factorization loss, $\mathcal{L}_{\mathrm{mf}}(\mathbf{F})$, where $\mathbf{F}_i = f_{\mathrm{mf}}(v_i)$. Then, for any label $\mathbf{y}$, there exists a linear classifier $\mathbf{B}^* \in \mathbb{R}^{c \times K}$ with norm $\|\mathbf{B}^*\|_F \leq 1/(1 - \lambda_K)$ such that*

$$\mathbb{E}_{v_i} \left[ \|\vec{y}_i - \mathbf{B}^* f_{\mathrm{mf}}^*(v_i)\|_2^2 \right] \leq \frac{\phi^{\mathbf{y}}}{\lambda_{K+1}}, \tag{26}$$

*where $\vec{y}_i$ is the one-hot embedding of the label of node $v_i$. The difference between labels of connected data points is measured by $\phi^{\mathbf{y}}$, $\phi^{\mathbf{y}} := \frac{1}{E} \sum_{v_i, v_j \in \mathcal{V}} \mathbf{A}_{ij} \cdot \mathbb{1}[y_i \neq y_j]$.*

*Proof of Theorem 2.* This proof is a direct summary on the established lemmas in previous section. By Lemma 2 and Lemma 4, we have,

$$\mathbb{E}_{v_i} \left[ \|\vec{y}_i - \mathbf{B}^* f_{\mathrm{gcl}}^*(v_i)\|_2^2 \right] \leq \frac{\phi^y}{\hat{\lambda}_{K+1}} \tag{27}$$

where $\hat{\lambda}_i$ is the $i$-th smallest eigenvalue of the Laplacian matrix $\widehat{\mathbf{L}}_{sym} = \mathbf{I} - \widehat{\mathbf{A}}_{sym}$. Note that $\phi^y$ in Lemma 4 equals $1 - h_{edge}$.

Then we apply Theorem 1 and Lemma 1 for $h_{edge}$ to conclude the proof:

$$\mathbb{E}_{v_i} \left[ \|\vec{y}_i - \mathbf{B}^* f_{\mathrm{gcl}}^*(v)\|_2^2 \right] \leq \frac{1 - h_{edge}}{\hat{\lambda}_{K+1}} \leq \frac{\bar{\phi} + \sqrt{\frac{\sigma_{\max}^2(\mathbf{W}^\top \mathbf{W}) \log(2N^2/\delta)}{2D^2 \|\mathbf{x}^2\|_{\psi_1}}}}{\hat{\lambda}_{K+1}}$$
$$= \frac{\bar{\phi}}{\hat{\lambda}_{K+1}} + \sqrt{\frac{\sigma_{\max}^2(\mathbf{W}^\top \mathbf{W}) \log(2N^2/\delta)}{2D^2 \|\mathbf{x}^2\|_{\psi_1} \hat{\lambda}_{K+1}^2}} \tag{28}$$

## C.5 INFLUENCE OF RANDOM INITIALIZATION

Recall that the concentration inequality for inner product of embedding in Lemma 1 is:

$$\left| \mathbf{Z}_i^\top \mathbf{Z}_j - \mathbb{E}[\mathbf{Z}_i^\top \mathbf{Z}_j] \right| \leq \sqrt{\frac{\sigma_{\max}^2(\mathbf{W}^\top \mathbf{W}) \log(2N^2/\delta)}{2D^2 \|\mathbf{x}^2\|_{\psi_1}}}$$

As shown by Lemma 1, the accuracy of using the graph encoder to obtain the similarity between two nodes is determined by the maximum singular value of the weight of the graph encoder. In other words, the quality of the transformed graph is guaranteed if we can be sure that the maximum singular value of the weight (of the graph encoder) is bounded. When initializing the network, we generally use Gaussian initialization, such as, Kaiming and Lecun initialization (He et al., 2015; LeCun et al., 2012). Then $\mathbf{W}^\top \mathbf{W}$ will be a Wishart matrix, and the limiting spectral density can be shown to be the Marchenko-Pastur distribution (Wigner, 1993), whose largest singular value is bounded when $K/F$ is bounded, where $\mathbf{W} \in \mathbb{R}^{F \times K}$. As a result, the quality of the transformed graph can be guaranteed with a high probability under condition that $K/F$ is bounded.

On the other hand, we assume that for node $i$, its neighbor's labels are independently sampled from a distribution $P(y_i)$. This means that the probability of an edge between two nodes can be fixed given labels. In this case, the degree scales linearly with number of nodes $N$. In the large graph limit, $D$ will tends to infinity, thus our bound can be tighter.

**Table 9:** Summary of graph augmentations used by representative GCL models. Multiple* denotes multiple augmentation methods including edge removing, edge adding, node dropping and subgraph induced by random walks.

| Method | Topology Aug. | Feature Aug. |
|---|---|---|
| DGI (Velickovic et al., 2019) | Node Shuffling | - |
| GMI (Peng et al., 2020) | Node Shuffling | - |
| MVGRL (Hassani & Khasahmadi, 2020) | Diffusion | - |
| GCC (Qiu et al., 2020) | Subgraph | - |
| GraphCL (You et al., 2020) | Multiple* | Feature Dropout |
| GRACE (Zhu et al., 2020d) | Edge Removing | Feature Masking |
| GCA (Zhu et al., 2021b) | Edge Removing | Feature Masking |
| BGRL (Grill et al., 2020) | Edge Removing | Feature Masking |

## D  AUGMENTATIONS IN EXISTING WORKS

## E  DISCUSSION

### E.1  CONNECTION WITH EXISTING OBSERVATIONS

In Lemma 1, we theoretically analyze the concentration property of aggregated features of graphs following the Graph Assumption (Section 4.1). The concentration property has also been empirically observed in some other recent works (Wang et al., 2022b; Trivedi et al., 2022). Specifically, in the Figure 1 of Wang et al. (2022b), a t-SNE visualization of node representations of the Amazon-Photo dataset shows that the representations obtained by a randomly initialized untrained SGC (Wu et al., 2019) model will concentrate to a certain area if they are from the same class. This observation also provides an explanation for the salient performance achieved by SP-GCL  on the Amazon-Photo, as shown in Table 1.

### E.2  LIMITATION AND FUTURE WORK

Our work is only the first step in understanding the possibility of single-pass graph contrastive learning and the corresponding performance guarantee on both the homophilic graph and heterophilic graph. There is still a lot of future work to be done. Below we indicate three questions that need to be addressed.

- Our theoretical framework is built upon the graph assumption (Section 4.1) in which we assume the neighbor pattern. The graph assumption includes the homophilic graph and the "benign" heterophilic graph. However, there could exist graphs in which the neighbor pattern is messy or arbitrarily distributed. It is still an open question to understand the graph contrastive learning on those graphs.
- Recently, several Graph neural networks (Zhu et al., 2020b; Chien et al., 2020; Luan et al., 2021; Li et al., 2022) have been proposed to work on heterophilic graphs. Whether further improvement can be achieved through combining with those GNNs is still an open question.
- To verify the effectiveness of our theoretical findings, we keep the implementation to be simple. Whether a better performance can be achieved by involving more complex techniques still need to be explored.

