# OpenReview forum: "Can Single-Pass Contrastive Learning Work for Both Homophilic and Heterophilic Graph?"
_ICLR.cc/2023/Conference — Submitted to ICLR 2023_

### Official Review · Reviewer_5Ey9 · 2022-10-24

**Confidence:** 4
**Clarity, Quality, Novelty And Reproducibility:** Please see above.
**Correctness:** 3
**Technical Novelty And Significance:** 2
**Empirical Novelty And Significance:** 2
**Recommendation:** 3

**Strength And Weaknesses:**

This paper is generally well organized, and the motivation of the proposed method is clearly described. The reviewer has the following concerns:

(i) Originality: My first concern is the novelty, and the reviewer feels the technical contributions are quite limited. For theoretical contribution, the theoretical analysis provided by the paper is quite similar to previous works. For instance, Lemma 1 (Concentration Property of Aggregated Features) has been proposed by work [1]; the inner product with the spectral norm in Lemma 1 is just a trivial extension of [1]. Lemma 2  and Theorem 2 (contrastive loss is equivalent to the matrix factorization) have also been proposed by [2]. The concept of the augmentation graph has been proposed by  [2]. Moreover, the proposed objective function (Single-Pass Graph Contrastive Loss in Eq. (9)) is the same as the spectral contrastive loss proposed by previous work [2]. The contribution of this paper is over-claimed.

(ii) Clarity. The design of the model is also unclear, and there is no clear connection between the theoretical results and the algorithm. For instance, the design of hard top-k positive and negative pairs and neighbor sampling are important but do not have theoretical justification. There are several assumptions on the node label/feature/neighborhood distributions. The non-linear operations in the GCN have also been dropped. Although theorem 1 is related to homophily and heterophily, why can the single-pass graph contrastive loss address the learning problem homophily in definition 1? What is the relation between Theorem 1 and single-pass graph contrastive loss? Current theorems can not prove the effectiveness of the proposed objective on homophily and heterophily graphs.

(iii) Quality: The experimental results are also not sufficient. Lack of some important ablation studies and qualitative analysis. For instance, what are the effects of top-k positive sampling and surrounding T-hop neighbors on the performance? To gain more insight into the behavior of the unsupervised method, it would also be better to visualize the embedding space learned.

[1] Is Homophily a Necessity for Graph Neural Networks. ICLR 2022.
[2] Provable Guarantees for Self-Supervised Deep Learning with Spectral Contrastive Loss. NeurIPS 2021.

**Summary Of The Paper:**

This paper studies the problem of conducting contrastive learning on graphs and introduce a single-pass graph contrastive learning framework for both homophilic and heterophilic graphs. The authors also try to provide some theoretical analysis to demonstrate that the design is effective. The experimental results show that the proposed objective outperforms existing graph contrastive learning works.

**Summary Of The Review:**

Overall, this paper's high-level idea and the objective function have been proposed by previous work [2]. The clarity and evaluation of this paper should also be improved.

---

> ### Author Response · Authors · 2022-11-12
> **Author Response to Reviewer 5Ey9**
>
> Thanks very much for your comments and suggestions. We would like to address your concerns as the following:
>
> 1. Technical contributions are quite limited.
>
> * The theoretical analysis is non-trivial and, in the graph domain, the result is important. The original spectral contrastive loss and its corresponding analysis (proposed in [2]) cannot be directly applied to the node-level graph contrastive learning. Not only because the spectral contrastive loss is designed under the (i.i.d.) image classification setting, but also because it requires a “good” image augmentation. In [2], a detailed discussion and assumption (Assumption 3.5 - "Labels are recoverable from augmentations") surrounding the image augmentation are given. Those assumptions induce the concept of **augmentation graph**. Instead, our analysis is augmentation-free and the analysis for our **transformed graph** is given.
> * Second, for the Concentration Property, we analyze the concentration property of the inner product and present it as a lemma (instead of overclaiming it as a new theorem) in our paper. Further,  Lemma 1 is for the development of theorem 1 which is able to justify the validity of selecting the positive nodes according to the inner product of hidden representation vectors.
> In all, the results and theoretical analysis obtained by [2] (i.i.d. and augmentation-based) cannot be trivially applied in our work (non-i.i.d. and augmentation-free).
>
>
> 2. The design of the model is also unclear.
> * We provide a figure about the architecture in Section 5 of the revision.
>
> 3. There is no clear connection between the theoretical results and the algorithm.
> * The connection: The theoretical results motivate the design of our algorithm. The loss actually employed to optimize the model (Equ.9) is an unbiased estimation of Equation.5 which is the focus of theoretical analysis.
> * Justification for the top-k positive selection: Lemma 1 and Theorem 1 justify the validity of selecting the positive nodes according to the inner product.
> * The Gaussian mixture modeling on the feature, the neighborhood patterns modeling on the graph structure, and the linearization on the graph neural network are commonly adopted in several recent works [3,4,5] for the theoretical analysis of GNNs. The high-level conclusions derived from their works still hold empirically for a wide range of real-world cases. In our case, we also empirically verify the derived Concentration Property(Section 6.3). We believe that a solid understanding of the behavior of simpler networks can improve our understanding of deep and complex ones.
>
>
>
> 4. Why can the single-pass graph contrastive loss address the learning problem homophily in definition 1?
> * The proposed SPGCL follows the heuristic "Pull node from same class together, and push node from different class away". Specifically, SPGCL constructs positive/negative pairs based on the aggregated node features and directly minimizes (maximizes) the distance of nodes of positive(negative) pairs.
> * As shown by Lemma 1, nodes with the same label tend to have the same embedding in expectation. The concentration property is invariant to the homophily degree. Thus, the procedure of positive pair construction (based on the aggregated feature) is independent of the homophily degree.
>
>
> 5. What is the relation between Theorem 1 and single-pass graph contrastive loss?
> * In Theorem 1, we theoretically show that the value of the inner product of the nodes from the same class is larger than the value of different classes, which justifies the feasibility of the construction of contrastive pairs by computing the inner product.

---

> > ### Author Response · Authors · 2022-11-12
> > **Author Response to Reviewer 5Ey9 (Cont.)**
> >
> >
> > 6. What are the effects of top-k positive sampling and surrounding T-hop neighbors on the performance?
> > * To scale up SPGCL on the large graph, we introduce the sampling mechanism as shown in Step.3 of Algorithm 1. To further study the hyperparameters involved in the sampling, we provide the ablation study about T and K in Appendix B.1 and B.2. We observe that the SPGCL is insensitive to the selection of T and K which indicates that the sampling mechanism adopted in this work is an effective way to scale SPGCL on large graphs.
> >
> > [1] Yao Ma, Xiaorui Liu, Neil Shah, and Jiliang Tang. Is homophily a necessity for graph neural networks? arXiv preprint arXiv:2106.06134, 2021.
> >
> > [2] Jeff Z HaoChen, Colin Wei, Adrien Gaidon, and Tengyu Ma. Provable guarantees for self-supervised deep learning with spectral contrastive loss. arXiv preprint arXiv:2106.04156, 2021.
> >
> > [3] Yash Deshpande, Subhabrata Sen, Andrea Montanari, and Elchanan Mossel. Contextual stochastic block models. Advances in Neural Information Processing Systems, 31, 2018.
> >
> > [4] Aseem Baranwal, Kimon Fountoulakis, and Aukosh Jagannath. Graph convolution for semi-supervised classification: Improved linear separability and out-of-distribution generalization. arXiv preprint arXiv:2102.06966, 2021
> >
> > [5] Yao Ma, Xiaorui Liu, Neil Shah, and Jiliang Tang. Is homophily a necessity for graph neural networks? arXiv preprint arXiv:2106.06134, 2021.

---

> ### Author Response · Authors · 2022-11-16
> **Friendly Reminder**
>
> Dear Reviewer 5Ey9:
>
> Thanks a lot for your efforts in reviewing this paper. We tried our best to address the mentioned concerns. Are there unclear explanations here? We could further clarify them.
>
> Best,
>
> Authors.

---

### Official Review · Reviewer_Lvdk · 2022-10-25

**Confidence:** 5
**Correctness:** 4
**Technical Novelty And Significance:** 4
**Empirical Novelty And Significance:** 4
**Recommendation:** 8

**Clarity, Quality, Novelty And Reproducibility:**

* Clarity. Very easy to follow and direct (without any syntactic sugar).
* Novelty. Theoretical underpinning and nice experimental results.
* Reproducibility. Code released with novel datasets.

**Details Of Ethics Concerns:**

Ok

**Strength And Weaknesses:**

* Strengths. The simplicity of single-pass contrastive learning with a nice theoretical background (concentration bounds).
* Weaknesses. K-hops are not really scalable (suggest using other measures such as learnable commute times to avoid OOM).

**Summary Of The Paper:**

Summary and main concern. Is a dual-pass necessary for computing contrastive loss even when heterophily is taken into account? The concentration analysis and the factorization interpretation are very interesting in answering this question. A single passs contrastive learning is implemented. Many benchmarks are explored in the experimental section.

The key to this insight is to embed nodes from the same class close and nodes from different classes far apart. In this regard, aggregated features as normal variables less to bound the variance (concentration lemma). Interestingly the bounds are governed by the singular values of the weights and also their correlation. As a result, nodes with similar labels have a similar embedding, and a contrastive loss can emerge from distinguishing close nodes from far away (in the sense of adjacency powers or k-hops).  Close nodes lead to positive edges (pairs) with in turn lead to a transformed graph that bridges graph contrastive learning with matrix factorization.

Experimental results are very impressive. The appendix is very informative and the code is released.

Suggestion. Using the concentration property explore the minimization o the entropy (can be a nice regularizer since the concavity of entropy)

**Summary Of The Review:**

A nice paper, very direct and out-standing.

---

> ### Author Response · Authors · 2022-11-12
> **Author Response to Reviewer Lvdk**
>
> Thank you very much for the recognition of our work! Exploring the minimization of the entropy from the view of concentration property is an interesting point, but it is out of the scope of this work. We would like to explore it in future works.
>
> To scale up SPGCL on the large graph, we adopt the T-hop neighborhood sampling. Besides, to study the effect of the selection of top-K and T-hop neighborhoods, we provide the ablation study in Append B.1 and B.2 of the revision. And we find that SPGCL is insensitive to the selection of K and T, which indicates that the sampling mechanism is a feasible way for SPGCL to scale up on large graphs. We would like to explore the learnable commute times, as you suggested, in future work.
>
> Thanks again for your comments and suggestions.

---

### Official Review · Reviewer_yTDn · 2022-11-03

**Confidence:** 4
**Correctness:** 3
**Technical Novelty And Significance:** 2
**Empirical Novelty And Significance:** 3
**Recommendation:** 5

**Clarity, Quality, Novelty And Reproducibility:**

This paper is over-all well written.

Please see the above comments about the concerns on the contributions.

**Strength And Weaknesses:**

Strengths:

1. Directly exploring the inner-product of hidden representations as the similarity to select positive samples is able to handle the heterophilic case, which is more reasonable than typical methods such as DGI and GMI where the positive samples are constructed from the neighbors.

2. While the proposed method is simple, the authors have tried to theoretically reveal the benefits of the proposed method. Although the conclusions from Lemma 1 to Theorem 1 are not that surprising, they somehow strictly explain the motivation of this paper.

3. Experiments are fully carried out, with sufficient comparisons with many baselines and the ablations. Even in certain cases, the performance is not as good as SOTA, the validity of the proposed method and stability of the training process are mostly supported.

Weaknesses:

1. The assumption that each node is generated IID is not very reasonable. This makes the conclusion of Lemma 1 irrelated to the graph structure of the input data. In graph learning, people are more interested in how the distribution of different node is related to each other, and prefer that the nodes are sampled from a non-IID distribution. Otherwise, Lemma 1 is also applicable to the case when using MLP (instead of GNN) for node hidden representation, which is not desirable. When considering the goal of this paper, Lemma 1 is also less informative in telling how the node homophily affects the hidden representation.

2. Lemma 2 and Theorem 1 are interesting, and make valuable results on how the contrastive loss in Eq. (5) is related to transformed graph, why performing contrastive learning helps node classification (Eq. 7), and how the edge homophily of the transformed graph affects the contrastive learning bound (Eq. 7). However, all these results are unable to justify the validity of selecting the positive nodes according to the inner product of hidden representation vectors, which is the central contribution of this paper. Lemma 2 and Theorem 1 hold regardless of how the transformed graph (the positive samples) is constructed. It is still unclear why the proposed theorems can support the benefit of the proposed single-pass contrastive learning against previous methods such as DGI and GMI. In other words, if we build the transformed graph by DGI or other sampling method, do Lemma 2 and Theorem 1 still hold? If so, what is the point they make?

I am happy if the authors can comment if I have missed something important.


Questions:

1. What is the theorem in (Ma et al. 2021)

2. It is suggested to indicate that F_i is Z_i in Lemma 2.

3. What is the relationship between Lemma 1 and other theorems?


**Summary Of The Paper:**

The authors propose a single-pass contrastive learning method for graph data, without the need of graph augmentation or encoder perturbation to generate contrastive samples. In particular, they select the positive samples based on the inner-product of the hidden representation for each central node. The method is quite simple. For evaluations on both homophilic and heterophilic graphs, the performance of the proposed method is sufficiently justified.

**Summary Of The Review:**

The biggest concern about this paper is that the theoretical results seem unable to support the benefit of the proposed contrastive learning method against other previous approaches.

---

> ### Author Response · Authors · 2022-11-12
> **Author Response to Reviewer yTDn**
>
> Thanks very much for the positive comments and constructive suggestions. We want to address your concerns and questions as the following:
>
> Concerns:
> 1. The assumption that each node is generated IID is not very reasonable.
>
> * Under the assumption of this work, the nodes are related to each other, given the graph structure. In other words, the nodes are non-$i.i.d.$. Specifically, the $(x_i, y_i)$ are non-$i.i.d.$ given the graph structure, because we make the assumption on the neighborhood pattern: for node $i$, its neighbor’s labels are sampled from a distribution $P(y_i)$. Note, the node features are conditional i.i.d., but this doesn't mean the node is i.i.d. Besides, the generation procedure of the node feature is commonly used. For example, in the well-known CSBM [4], the same feature generation assumption is adopted.
> * Besides, the conclusion of Lemma 1 is related to the graph structure of the input data but is irrelated to the homophily degree of the graph data. Lemma 1 focuses on the node features obtained through neighborhood aggregation, in which the message passing is determined by the graph structure.
>
>
>      Hope our explanation is able to address your concerns about the data assumption in this work.
>
>
> 2. The validity of selecting the positive nodes according to the inner product of hidden representation vectors
>
> * Lemma 1 and Theorem 1 justify the validity of selecting the positive nodes according to the inner product. Lemma 1 (concentration property) illustrates that the representation of nodes from the same class tends to concentrate on their class center. Intuitively, for each node, the closest nodes (selected according to the inner product) are more possible belong to the same class. Furthermore, in Theorem 1, we theoretically show that the value of the inner product of the nodes from the same class is larger than the value of a different class.
> * Besides, we also empirically verify the feasibility of the procedure of positive node selection. As shown in Table 4b, the edge homophily of all transformed graphs is larger than the original graph. The result justifies the validity of selecting the positive nodes according to the inner product with real-world graph data.
>
>
> 3. It is still unclear why the proposed theorems can support the benefit of the proposed single-pass contrastive learning against previous methods such as DGI and GMI. In other words, if we build the transformed graph by DGI or other sampling methods, do Lemma 2 and Theorem 1 still hold? If so, what is the point they make?
>
> * Unifying all GCL methods into one theoretical framework and comparing them is out of the scope of this work. We would leave it in future work.
> * Recent work [2] show that previous GCL methods, such as DGI and GMI, will only capture the low-frequency information and ignore the high-frequency information which is important for the node classification task over heterophilic graphs [3].
> * Against previous methods, our analysis theoretically shows that SPGCL is able to work well on both homophilic and heterophilic graphs, which supports the benefit of the proposed single-pass contrastive learning against previous methods such as DGI and GMI.
> * Constructing a transformed graph by DGI cannot lead us to obtain a meaningful conclusion. First, the concept of "transformed graph" is only used to help us better illustrate the proof of the performance guarantee for the single-pass graph contrastive loss (Theorem 2). Other GCL methods don't use the loss, so we cannot reach a conclusion by analyzing the transformed graph constructed by other methods, such as DGI and GMI.

---

> > ### Author Response · Authors · 2022-11-12
> > **Author Response to Reviewer yTDn (Cont.)**
> >
> >
> > Questions:
> > 1. What is the theorem in Ma et al. 2021?
> > * The Theorem 1 of Ma et al.[1] aims to show that, in expectation, all nodes with the same label have the same embedding and the distance between the output embedding of a node and its expectation is small with a high probability.
> >
> >
> > 2. It is suggested to indicate that F_i is Z_i in Lemma 2.
> > * They are not (always) the same, therefore is inappropriate to say "F_i is Z_i" in Lemma 2 directly. The optimal solution of matrix factorization is $\mathbf{F}^*$ which is equal to the optimal solution of the graph contrastive loss. And the connection and equivalence between $\mathbf{F}$ and $\mathbf{Z}$ are analyzed and discussed carefully in Appendix B.2.
> >
> >
> > 3. What is the relationship between Lemma 1 and other theorems?
> > * Lemma 1 qualitatively analyzes the concentration property of aggregated features and Theorem 1 provides a quantitative result of the gap between the inner product of the nodes from the same class or different class. Besides, the derivation of Theorem 2 needs Lemma 1 as shown in Appendix C.4.
> >
> >
> >
> > [1] Yao Ma, Xiaorui Liu, Neil Shah, and Jiliang Tang. Is homophily a necessity for graph neural networks? arXiv preprint arXiv:2106.06134, 2021.
> >
> > [2] Nian Liu, Xiao Wang, Deyu Bo, Chuan Shi, and Jian Pei. Revisiting graph contrastive learning from the perspective of graph spectrum.
> >
> > [3] Deyu Bo, Xiao Wang, Chuan Shi, and Huawei Shen. Beyond low-frequency information in graph convolutional networks. arXiv preprint arXiv:2101.00797, 2021.
> >
> > [4] Yash Deshpande, Subhabrata Sen, Andrea Montanari, and Elchanan Mossel. Contextual stochastic block models. In NeurIPS, 2018.

---

> ### Author Response · Authors · 2022-11-16
> **Friendly Reminder**
>
> Dear Reviewer yTDn:
>
> Thanks a lot for your efforts in reviewing this paper. We tried our best to address the mentioned concerns. Are there unclear explanations here? We could further clarify them.
>
> Best,
>
> Authors.

---

### Official Review · Reviewer_RpbG · 2022-11-03

**Confidence:** 3
**Correctness:** 3
**Technical Novelty And Significance:** 2
**Empirical Novelty And Significance:** 2
**Recommendation:** 3

**Clarity, Quality, Novelty And Reproducibility:**

The new idea in contrastive learning is provided, but it lacks detailed comparisons with the literature under the heterophily settings as mentioned before.

**Strength And Weaknesses:**

Pros:
1. The proposed single-pass contrastive loss is interesting and effective for both two kinds of graphs.
2. The model performance can be guaranteed under a set of assumptions.


Cons:

Compared with the existing contrastive learning methods, the advantages of single-pass loss are unclear.
Based on the assumptions mentioned in Section 4.1, the Concentration Property can be guaranteed for both two graphs. Then, what are the challenges to applying contrastive learning methods on the heterophilous graphs? And how the proposed single-pass loss solved this problem? What are the advantages? Besides,
In the experiments, it would be better to show the influence of $K_{pos}$ in Section 6.3.


**Summary Of The Paper:**

In summary, this paper provides one single-pass contrastive learning method which can work for both homophiles and heterophilous graphs. One single-pass contrastive loss is designed based on the concentration property, and its minimizer is equivalent to MF over the transformed graph.



**Summary Of The Review:**

It is interesting to designing the single-pass loss for contrastive learning, while it is not well motivated when applying them under the heterophily settings. Then, the contribution compared with the existing methods is limited.

---

> ### Author Response · Authors · 2022-11-12
> **Author Response to Reviewer RpbG**
>
>
> Thanks very much for your constructive comments and questions. We would like to address your concerns as the following:
>
>
> 1. What are the challenges to applying contrastive learning methods on the heterophilic graphs?
>
> * With empirical observation and theoretical analysis, recent work [1] pointed out that, in the previous graph contrastive learning methods, the contrastive pairs differ more in high-frequency components than in low-frequency ones. And the employed contrastive loss is essentially learning representations invariant to high-frequency components and capturing the low-frequency information, which is helpful for node classification on homophilic graphs. However, as shown by Bo et al. [2], high-frequency information is more useful for heterophilic graphs, which limits the success of previous contrastive learning methods on heterophilic graphs.
>
>
> 2. How the proposed single-pass loss solved this problem? What are the advantages?
> * The proposed SPGCL follows the heuristic "Pull node from same class together, and push node from different class away". Specifically, SPGCL constructs positive/negative pairs based on the aggregated node features and directly minimizes (maximizes) the distance of nodes of positive(negative) pairs.
>
>   *Advantages:*
>   * Through the analysis (Lemma 1), we show that the feature concentration property and the construction of positive/negative pairs based on the aggregated features is intuitively reasonable.
>   * Theoretically, we justify the validity of selecting the positive nodes according to the inner product of hidden representation vectors (Theorem 1). Further, we prove that by minimizing the loss (Equ.9), the node classification performance is guaranteed.
>   * Empirically, compared with other GCL methods, our method (SPGCL) is able to perform well on both homophilic and heterophilic graphs with a significant reduction of computational overhead.
>
>
> 3. It would be better to show the influence of K-pos in Section 6.3.
> * We provide the ablation study of the effect of top-K in Appendix B.1. Besides, because the selection of T-hop neighbor is also involved in the batch sampling, we show the effect of T-hop neighbors on the model performance in Appendix B.2. We observed that SP-GCL is insensitive to the selection of K and T. For more details, we would like to refer the reviewer to Appendix B.1 and B.2 of the revision.
>
>
> 4. Lacks detailed comparisons with the literature under the heterophily settings.
> * We describe the limitation of the previous methods under the heterophily setting in the introduction section of the revision. Besides, careful empirical comparisons are given in Tables 1 & 2, which justify the effectiveness of the proposed method.
>
>
> [1] Liu, Nian, Xiao Wang, Deyu Bo, Chuan Shi, and Jian Pei. Revisiting Graph Contrastive Learning from the Perspective of Graph Spectrum. arXiv preprint arXiv:2210.02330 (2022).
>
> [2] Deyu Bo, Xiao Wang, Chuan Shi, and Huawei Shen. Beyond low-frequency information in graph convolutional networks. arXiv preprint arXiv:2101.00797, 2021.

---

> ### Author Response · Authors · 2022-11-16
> **Friendly Reminder**
>
> Dear Reviewer RpbG:
>
> Thanks a lot for your efforts in reviewing this paper. We tried our best to address the mentioned concerns. Are there unclear explanations here? We could further clarify them.
>
> Best,
>
> Authors.

---

### Public Comment · ~Benedek_Andras_Rozemberczki1 · 2022-11-05
**Misattribution of multiple datasets**

The paper misattributed the authorship of the Chameleons and Squirrels datasets. These datasets were proposed in this ICLR submission:

https://openreview.net/forum?id=HJxiMAVtPH

The Pei et al. paper cited by the authors took the Squirrel and Chameleons datasets and used those for benchmarking, but had nothing to do with the creation of the datasets. The correct citation for the paper which proposed the datasets is:

```bibtex
>@article{musae,
          author = {Rozemberczki, Benedek and Allen, Carl and Sarkar, Rik},
          title = {{Multi-Scale Attributed Node Embedding}},
          journal = {Journal of Complex Networks},
          volume = {9},
          number = {2},
          year = {2021},
}
```

The Twitch-Gamers dataset was proposed in this paper:

```bibtex

>@misc{rozemberczki2021twitch,
       title = {Twitch Gamers: a Dataset for Evaluating Proximity Preserving and Structural Role-based Node Embeddings},
       author = {Benedek Rozemberczki and Rik Sarkar},
       year = {2021},
       eprint = {2101.03091},
       archivePrefix = {arXiv},
       primaryClass = {cs.SI}
       }

```

---

> ### Author Response · Authors · 2022-11-12
> **Author Response**
>
> Hello Benedek,
>
> Thanks for your interest in our work! We have corrected the citation about the Squirrel, Chameleons, and Twitch Gamers in the revision.
>
> Thanks again.

---

### Decision · Program_Chairs · 2023-01-20

**Decision:**

Reject

**Justification For Why Not Higher Score:**

The position of this submission needs carefully revised, the connections between "single-pass", "homophilic and heterophilic", and "theoretical results" should be trimmed.

**Justification For Why Not Lower Score:**

N/A

**Metareview: Summary, Strengths And Weaknesses:**

This paper proposes a new method for contrastive learning on graphs, which contains two unique properties, i.e., one forward-pass and work for both homophilic and heterophilic graphs. To motivate the design of the proposed method, the authors theoretically study similarities between aggregated features. Then, authors develop a single-pass contrastive learning method with performance guarantee. Extensive experiments have also been done, which show the proposed method can achieve SOTA performance.

Weakness
- Theoretical results lack proper justification. Certain important design choices (e.g., positive and negative pairs and neighbor sampling) are not analyzed.
- Motivation is not clearly stated. It is not clear why the obtained theoretical results can support the two main designs of the method.

Strength
- Extensive experiments have been performed.
- Method is supported with theoretical analysis.